# Large-Scale Differentiable Causal Discovery of Factor Graphs

**Romain Lopez**[1,2], **Jan-Christian Hütter**[1],
**Jonathan K. Pritchard**[2,3,†], **Aviv Regev**[1,†]

[1] Division of Research and Early Development, Genentech
`{lopez.romain, huettej1, regeva}@gene.com`

[2] Department of Genetics, Stanford University
`pritch@stanford.edu`

[3] Department of Biology, Stanford University

[†] These authors contributed equally to this work.

## Abstract

A common theme in causal inference is learning causal relationships between observed variables, also known as causal discovery. This is usually a daunting task, given the large number of candidate causal graphs and the combinatorial nature of the search space. Perhaps for this reason, most research has so far focused on relatively small causal graphs, with up to hundreds of nodes. However, recent advances in fields like biology enable generating experimental data sets with thousands of interventions followed by rich profiling of thousands of variables, raising the opportunity and urgent need for large causal graph models. Here, we introduce the notion of factor directed acyclic graphs ($f$-DAGs) as a way to restrict the search space to non-linear low-rank causal interaction models. Combining this novel structural assumption with recent advances that bridge the gap between causal discovery and continuous optimization, we achieve causal discovery on thousands of variables. Additionally, as a model for the impact of statistical noise on this estimation procedure, we study a model of edge perturbations of the $f$-DAG skeleton based on random graphs and quantify the effect of such perturbations on the $f$-DAG rank. This theoretical analysis suggests that the set of candidate $f$-DAGs is much smaller than the whole DAG space and thus may be more suitable as a search space in the high-dimensional regime where the underlying skeleton is hard to assess. We propose Differentiable Causal Discovery of Factor Graphs (DCD-FG), a scalable implementation of $f$-DAG constrained causal discovery for high-dimensional interventional data. DCD-FG uses a Gaussian non-linear low-rank structural equation model and shows significant improvements compared to state-of-the-art methods in both simulations as well as a recent large-scale single-cell RNA sequencing data set with hundreds of genetic interventions.

## 1 Introduction

Characterizing causal dependencies between variables is a fundamental problem in science [1, 2, 3]. Such relationships are often described via a directed acyclic graph (DAG) where nodes represent variables and directed edges encode causal links between pairs of variables. One may then consider a

36th Conference on Neural Information Processing Systems (NeurIPS 2022).

Table 1: Comparison with related work. $d$ denotes the number of features, $m$ denotes the number of learned factors. Because $m$ remains low (e.g., 20), we have $m \ll d$. $^\dagger$additive cubic model only.

| Related work | non-linear link function | likelihood eval. complexity | DAG penalty eval. complexity | intervention model |
|---|---|---|---|---|
| **NO-TEARS** [19] | ✗ | $O(d^2)$ | $O(d^3)$ | ✗ |
| **NO-BEARS** [26] | ✓$^\dagger$ | $O(d^2)$ | $O(d^2)$ | ✗ |
| **NO-TEARS-LR** [20] | ✗ | $O(md)$ | $O(d^3)$ | ✗ |
| **DCDI** [6] | ✓ | $O(d^2)$ | $O(d^3)$ | ✓ |
| **DCD-FG** (this work) | ✓ | $O(md)$ | $O(md)$ | ✓ |

set of conditional probability distributions in addition to the DAG to define a causal graphical model (CGM) [4]. The graphical model is referred to as "causal" because it provides a clear semantic for defining the effect of interventions on the joint likelihood [5].

The combinatorial nature of the set of DAGs and its size make inference of causal structure from data a hard problem. Indeed, the number of DAGs (or the number of permutations) grows super exponentially in the number of nodes, limiting the practical application of most methods to tens of nodes [6, 7, 8]. This highlights a disparity between the development of the field and emerging data from real-world problems in finance [9, 10], neuroscience [11, 12] and high-throughput biology [13, 14, 15, 16]. For example, experimental advances in biology have enabled the generation of data sets where expression profiles of thousands of RNA transcripts are measured in hundreds of thousands of cells following genetic interventions in each of hundreds or thousands of different genes.

To reduce this computational burden, a promising direction is to limit the complexity of the search space. This idea has become commonplace at the interface of statistics and optimization, where sparsity [17] as well as low-rank assumptions [18] are often exploited in machine learning algorithms. Although sparsity is widely used in causal structure learning [7, 19], the use of low-rank constraints in this specific setting remains largely under-explored, with a few notable exceptions [20, 21]. This may be due to the fact that such an assumption may be hard to incorporate into common methods for DAG learning that rely on graphical constraints or on permutations.

Recently, the NOTEARS methodology [19] introduced a continuous relaxation of DAG learning, effectively closing the gap between causal structure learning and continuous optimization. NOTEARS facilitates incorporating more flexible structural assumptions into DAG learning, such as neural network parametrizations of the conditional probability distributions. Although NOTEARS has been the subject of several follow-up works [22, 23, 24, 25, 26], including learning from interventional data [6], several challenges remain. First, the likelihood function usually decomposes as a product of local likelihoods for each node, and existing options for those local models are either fast to fit but likely underfitting the data (linear models [19]) or are computationally expensive and prone to overfitting (a neural network taking as input all of the other nodes [6]). Second, most methods rely on a differentiable penalty for acyclicity whose evaluation has cubic complexity in the number of nodes. As a result, these methods are impractical for graphs with more than a hundred nodes.

In this work, we investigate both challenges and propose a methodology for large-scale discovery of causal structure and prediction of unseen interventions that scales to millions of samples and thousands of nodes. Our key idea is to limit the search space to what we refer to as factor directed acyclic graphs ($f$-DAGs), a class of low-rank graphs defined in Section 3.1. This constraint assumes that many nodes share similar sets of parents and children, which is the case in scale-free topologies [20] and many biological systems, where genes act together in programs [3, 27, 28]. Based on this class of graphs, we design a flexible model and a scalable inference procedure (Table 1) that we refer to as Differentiable Causal Discovery of Factor Graphs (DCD-FG).

After introducing the necessary background (Section 2), we define the class of $f$-DAGs (Section 3) and draw connections to several flavors of matrix factorization. In particular, we show connections between the number of factors in an $f$-DAG and the Boolean rank [29] of its adjacency matrix. We exploit these connections to present a scalable acyclicity score with linear complexity in the number of observed variables. Finally, we characterize the identifiability of these graphs under an Erdős-Rényi random graph model and prove the instability of the Boolean rank under edge perturbations. The latter analysis highlights that restricting inference to $f$-DAGs is more efficient in the high-

dimensional causal discovery regime. Then, we posit a flexible class of likelihood models as well as a scalable inference method for our DCD-FG framework (Section 4). Finally, we present runtime experiments, simulation studies, and a case study on single-cell RNA sequencing data with hundreds of genetic interventions (Section 5). In this last challenging instance of interventional data with high-dimensional measurements, we show that our framework outperforms current state-of-the-art causal discovery approaches for predicting the effect of held-out interventions.

## 2 Background

Our work builds upon continuous relaxations of the causal structure learning problem. We therefore first briefly introduce causal graphical models and then how the inference problem is solved with a gradient-based optimization framework.

### 2.1 Causal Graphical Models

Following the framework introduced in [5], let $X = (X_1, \ldots, X_d)$ denote a set of random variables with a joint probability distribution $P$. Let $G = (V, E)$ be a DAG where each vertex $v_i \in V$ is associated with a random variable $X_i$ and each edge $(v_i, v_j)$ represents a causal relationship from $X_i$ to $X_j$. A correspondence between the graph and the probability distribution $P$ is obtained via the factorization condition

$$P(X_1, \ldots, X_d) = \prod_{j=1}^{d} P(X_j \mid X_{\pi(j)}), \tag{1}$$

where $X_{\pi(j)}$ denotes the vector of random variables formed by all parents of node $v_j$ in $G$. We refer to a pair $(P, G)$ that satisfies the factorization condition as a causal graphical model (CGM).

Unlike classical graphical models, CGMs provide a principled way to model interventions [30]. With interventional data, each observation is measured under a specific regime $k$ with interventional joint density $P^{(k)}$. For each regime, a subset of the target variables $\mathcal{I}_k$ is subject to intervention. Each of these interventions affects the relationship between the target node and its parents, by altering the conditional distribution. In the case of perfect interventions [8], the dependency of intervened upon nodes from their parents is removed, and the interventional joint density $P^{(k)}$ becomes

$$P^{(k)} = \prod_{j \notin \mathcal{I}_k} P(X_j \mid X_{\pi(j)}) \prod_{j \in \mathcal{I}_k} P^{(k)}(X_j), \tag{2}$$

where $P^{(k)}$ models the effect of the perfect (stochastic) interventions on each targeted feature of $\mathcal{I}_k$.

### 2.2 Differentiable Causal Structure Learning

A significant challenge is to identify and learn CGMs from data when the causal relationships between features are not known beforehand. Several methods have been proposed for this problem, such as constraint-based methods (e.g., the PC algorithm [8]), score-based methods (e.g., GSP [31]), and their extensions for modeling interventions (e.g., IGSP [7]), all reviewed in [32].

The NOTEARS methodology [19] proposes to solve a continuous relaxation of the optimization problem of score-based methods, which in the case of observational data can be written as

$$\min_{\mathbf{W}} \frac{1}{n} \sum_{i=1}^{n} \left\| X^i - \mathbf{W} X^i \right\|_2^2 + \lambda \left\| \mathbf{W} \right\|_1 \quad \text{such that} \quad \text{Tr} \exp\{\mathbf{W} \circ \mathbf{W}\} = d, \tag{3}$$

where $n$ denotes the total number of i.i.d. observations, $X^i \in \mathbb{R}^d$ denotes the $i$-th observation of $X$, $\circ$ denotes the Hadamard product, $\mathbf{W} \in \mathbb{R}^{d \times d}$ denotes the parameters of a linear Gaussian conditional distribution, assuming equal variance for all nodes, and $\lambda > 0$ denotes a tuning parameter. The search space is restricted to DAGs by enforcing that the trace of the exponential (tr-exp) of $\mathbf{W} \circ \mathbf{W}$ is equal to its dimension $d$. Importantly, evaluating the tr-exp of $\mathbf{W} \circ \mathbf{W}$ as well its gradient with respect to $\mathbf{W}$ has $O(d^3)$ time complexity, which makes it potentially prohibitive for large-scale applications.

This fundamental work has been subject to multiple extensions, of which the most relevant ones are listed in Table 1. First, different models of relationships between variables were introduced with

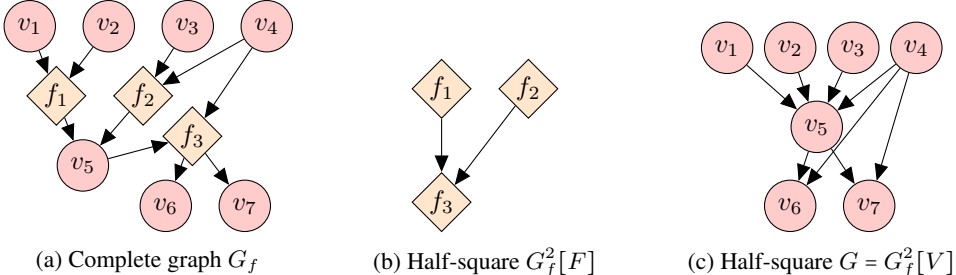

(a) Complete graph $G_f$      (b) Half-square $G_f^2[F]$      (c) Half-square $G = G_f^2[V]$

Figure 1: A factor graph and its induced half-square graphs. Red circles represent variable nodes. Orange diamonds represent factor nodes.

the aim of better fitting the data, including several non-linear models, such as neural networks [6, 22, 24, 33, 23]. Because these methods use one neural network per conditional distribution (or one deep sigmoidal flow), they are computationally expensive, as emphasized in [6]. The additive cubic model proposed by NOBEARS [26] does not have these scalability issues, but may be of limited flexibility. To reduce the number of parameters, graph neural networks have been proposed [25], but that architecture cannot be easily applied to the case of interventional data. In addition, NOTEARS-LR [20] proposes a low-rank decomposition of the linear model from NOTEARS but did not consider that the computational complexity could become a bottleneck for large graphs.

Second, studies also investigated variants of the acyclicity penalty. [25] proposes a matrix power variant of the trace exponential criterion, for numerical stability. NOBEARS [26] uses an algebraic characterization of acyclicity based on the spectral radius of the adjacency matrix, that can be approximated in $O(d^2)$. LoRAM [21] exploits a low-rank assumption to obtain a $O(d^2 m)$ acyclicity score, where $m$ denotes the rank. However, the proposed framework is tailored to projections of DAG into a low-rank space but not to causal discovery learning.

## 3    Factor Directed Acyclic Graphs

Next, we introduce factor directed acyclic graphs ($f$-DAGs) and draw connections to low-rank matrix factorizations. Then, we motivate their use in causal inference by studying the effect of edge perturbation on their rank. Additionally, we provide two differentiable acyclicity penalties computable in linear time in the number of nodes for a fixed set of factors by exploiting the low-rank structure.

### 3.1    Definitions and relationship to low-rank decomposition

Factor graphs, especially undirected ones, are commonly used in the graphical models literature to factorize probability distributions and describe the complexity of message passing algorithms [34]. In this work, we use them to construct causal graphs over features. In addition to the set of feature vertices $V = \{v_1, \ldots, v_d\}$, we consider a set of factor vertices $F = \{f_1, \ldots, f_m\}$, for $m \in \mathbb{N}$. When the edge set $E$ links only vertices of different types, the graph $G_f = (V, F, E)$ is a bipartite graph and we refer to it as a factor directed graph ($f$-DiGraph). If $G_f$ additionally does not contain cycles, we call it an $f$-DAG. $G_f$ canonically induces two half-square graphs $G_f^2[V]$ and $G_f^2[F]$, defined by drawing an edge between vertices of $V$ (or $F$) of distance exactly two in $G_f$ (see example in Figure 1). Unless otherwise mentioned, we will refer to $G = G_f^2[V]$ as the half-square of $G_f$ over vertices. We define the set of half-square graphs of $f$-DiGraphs with $d$ variables and $m$ factors as

$$\mathcal{G}_d^m = \left\{ G = G_f^2[V] \mid G_f = (V, F, E) \text{ is a factor directed graph and } |F| = m \right\}. \quad (4)$$

Intriguingly, the set $\mathcal{G}_d^m$ may be identified as the set of matrices with Boolean rank [29] of at most $m$.

**Proposition 1** (Bounded Boolean rank of half-square adjacency matrix)**.** *For a factor graph $G_f = (V, F, E)$, let $\mathbf{U} \in \{0,1\}^{d \times m}$ (resp. $\mathbf{V} \in \{0,1\}^{m \times d}$) be the binary matrix encoding the presence or absence of edges directed towards factor nodes (resp. variable nodes), according to $E$. The adjacency matrix $\mathcal{A}(G)$ of the half-square graph $G$ may be decomposed as $\mathcal{A}(G) = \mathbf{U} \diamond \mathbf{V}$ where $\diamond$ denotes the Boolean matrix product, $(\mathbf{U} \diamond \mathbf{V})_{ij} = \bigvee_{k=1}^m \mathbf{U}_{ik} \wedge \mathbf{V}_{kj}$, $i, j \in [d]$. Consequently, $\mathcal{A}(G)$*

*has Boolean rank bounded above by $m$. Conversely, every adjacency matrix over the feature nodes with Boolean rank bounded by $m$ can be written as half-square of an $f$-DiGraph with $m$ factors.*

This result, proven in Appendix A, establishes a connection between inference of causal $f$-DAGs and Boolean matrix factorization. Additionally, it is easy to notice that the (integer-valued) matrix product $\mathbf{UV}$ provides a valid (weighted) adjacency matrix for $G$. In this work, we show that the Boolean decomposition is suited for theoretical analysis of the method, while the linear one is useful for efficient algorithm design. We further note that this proposed class of graphs is smaller than the one arising from adjacency matrices with (unconstrained) linear matrix rank bounded by $m$, which we detail further in Appendix A.4.

### 3.2 Statistical Properties of Random Causal Factor Graphs

An important theoretical question pertains to the assumptions necessary for identification of acyclic graphs in $\mathcal{G}_d^m$ (which we denote by $\mathcal{D}_d^m$) from data. Under the classical set of assumptions (causal sufficiency, causal Markov property and faithfulness), we may identify the causal DAG from observational data only up to its Markov equivalence class (MEC) [35]. However, graphs in $\mathcal{D}_d^m$ can have many v-structures (emerging from having several feature parents), potentially making them identifiable. Indeed, we prove in Appendix B that under an adapted Erdős-Rényi random graph model [36] over $\mathcal{D}_d^m$, graphs are identifiable with high probability (i.e., their MEC is reduced to one graph). We also verified this with simulations (Figure 2A).

Although valid, the previous result may be disconnected from real-world applications in high-dimensional regimes for which stronger assumptions are required (e.g., strong faithfulness [37]), but rarely hold [38]. More concretely, we expect errors in the estimated skeleton of the feature node graph. As a toy model of these errors, we assume the true causal graph $G$ is in $\mathcal{G}_d^m$ and apply a stochastic edge perturbation operator $\Lambda$ that randomly removes or adds one edge to $G$. While there are no general rules as to how identifiability is changed under these perturbations [39] (i.e., the size of the MEC could increase or decrease), we show that the Boolean rank of the graph strictly increases with high probability:

**Theorem 1** (Boolean rank instability for edge perturbation). *Let $G \in \mathcal{G}_d^m$ be sampled according to an Erdős-Rényi random directed graph model. The probability that adding or removing an edge increases the Boolean rank is arbitrarily high for large $d$:*

$$\mathbb{P}\left(Rank_{\mathcal{B}}\left(\Lambda(G)\right) > Rank_{\mathcal{B}}\left(G\right)\right) \geq 1 - \alpha q^d, \tag{5}$$

*where $Rank_{\mathcal{B}}$ denotes the Boolean rank, and $\alpha > 0$ and $q \in (0,1)$ depend on $m$ and the parameters of the random graph model.*

Precise definitions of the random graph model and proof appear in Appendix C. This result, analogous to the upper semicontinuity of the matrix rank, suggests that in noisy settings, the skeleton of graphs in $\mathcal{D}_d^m$ may be more easily recoverable compared to arbitrary DAGs. We verified those results with simulations (Figure 2B) for small graphs using a mixed integer linear programming approach [40]. We performed a larger-scale analysis by randomly perturbing a fraction $q$ of the edges in the graph and reporting the resulting matrix rank[1] of the adjacency matrix for larger graphs (Figure 2C).

### 3.3 Characterizing Acyclicity of Factor Graphs

We start by relating the acyclicity of a $f$-DiGraph with the one of its induced half-squares. A simple graphical argument is enough to show that acyclicity for a $f$-DiGraph need only be enforced on the smaller of its half-square graphs.

**Lemma 1** (Induced acyclicity). *Let $G = (V, F, E)$ be an $f$-DiGraph. Then,*

$$G_f \text{ is acyclic } \Leftrightarrow G = G_f^2[V] \text{ is acyclic } \Leftrightarrow G_f^2[F] \text{ is acyclic.} \tag{6}$$

The proof can be found in Appendix D. We further note that the matrix $\mathbf{UV}$ ($\mathbf{VU}$, resp.) counts the number of paths between two nodes in $V$ ($F$, resp.) and is thus a valid (weighted) adjacency matrix

---

[1]Using the matrix rank is only a heuristic approximation of the quantitative increase in complexity of the underlying matrix, because in general it provides neither an upper nor a lower bound on the Boolean rank [41].

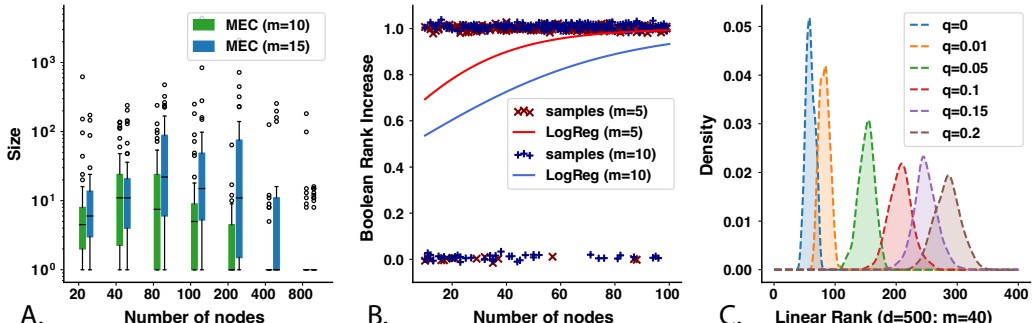

Figure 2: Properties of simulated Erdős-Rényi random $f$-DiGraphs. (A) Size of MEC of the simulated half-square graph with varying $m$ and $d$. (B) Probability of increase of Boolean rank after edge perturbation (Theorem 1). A single point denotes the result of an experiment, the solid line is the probability estimated by logistic regression. (C) Matrix rank change after multiple random edge perturbations.

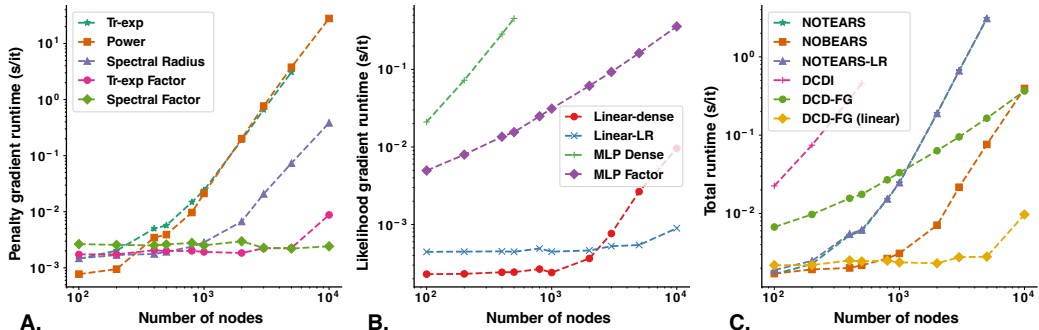

Figure 3: Runtime analysis. Time for gradient calculation of likelihood (A), penalty (B) or their sum (C) for different variants of optimization-based DAG inference method (on one NVIDIA Tesla T4 GPU with 15Gb of RAM). We selected a batch size of 128 datapoints and a number of $m = 40$ factors for all experiments. If a value is non-reported, a memory error was raised at runtime. NOTEARS and NOTEARS-LR have almost identical runtime in this analysis.

for $G$ ($G_f^2[F]$, resp.). As a result, we may characterize acyclicity by applying the tr-exp penalty to the matrix $\mathbf{VU}$ in time $O(m^3 + m^2 d)$ compared to the $O(d^3 + md^2)$ steps needed for evaluating the penalty on $\mathbf{UV}$. Alternatively, we may use the spectral radius of the adjacency matrix as an acyclicity score [26] that can be approximately computed in $O(Tmd)$ steps, with $T$ iterations of the power method between each gradient step (details in Appendix D). The resulting computational gains are showcased in Figure 3A. For small $m$, both variants have a similar runtime.

## 4 Differentiable Discovery of Causal Factor Graphs

To define a CGM, we couple the $f$-DiGraph with a likelihood model. We follow recent work that partitions the parameter space into conditional distribution parameters $\Theta$, and parameters $\Phi$ encoding the causal graph [6]. In particular, let us assume we have at our disposal a parameterized distribution $\mathbf{M}(\Phi) = [\mathbf{U}(\Phi), \mathbf{V}(\Phi)]$ over adjacency matrices of $f$-DiGraphs. The score function, assuming perfect interventions, is defined as:

$$\mathcal{S}(\Phi, \Theta) = \mathbb{E}_{\mathbf{M}' \sim \mathbf{M}(\Phi)} \left[ \sum_{k=1}^{K} \mathbb{E}_{X \sim P_{\text{data}}^{(k)}} \sum_{j \notin \mathcal{I}_k} \log p_\Theta^j(X_j; \mathbf{M}'_j, X_{-j}) \right] - \lambda \left\| \mathbb{E}\left[\mathbf{M}(\Phi)\right] \right\|_1, \quad (7)$$

where $P_{\text{data}}^{(k)}$ denotes the distribution of data points $X$ under regime $k$, $p_\Theta^j$ denotes a density model for feature $X_j$, conditioned on all other features $X_{-j}$ that are parents of the feature $j$ according to the

sampled matrix $\mathbf{M}'_j$. We optimize the score function $\mathcal{S}$ under an acyclicity constraint,

$$\max_{\Phi,\Theta} \mathcal{S}(\Phi,\Theta) \text{ such that } \mathcal{C}(\mathbb{E}[\mathbf{M}(\Phi)]) = 0, \tag{8}$$

where $\mathcal{C}$ may correspond to either the spectral radius, or the tr-exp characterization of acyclicity. Numerically, first-order optimization techniques with reparameterized gradients and the augmented Lagrangian method are used to solve problem (8). We outline here some key features of DCD-FG, and provide the complete implementation details in Appendix E.

**Differentiable Sampling of Factor Graphs**  A first important challenge specific to our work is constructing a density $\mathbf{M}(\Phi)$ over $f$-DiGraphs. The DCDI framework [6], and a few earlier methods [23, 42], parameterize the set of adjacency matrices with entry-wise Gumbel-sigmoid [43] samples, and zeros in the diagonal entries. Naively applying this parameterization for sampling matrices $\mathbf{M} = [\mathbf{U}, \mathbf{V}]$ causes the induced feature graphs to have a large number of self-loops, i.e., edges of the form $(v, v)$ that we found to be detrimental to the performance of the model. To circumvent this issue, we propose an alternative model in which the matrices $\mathbf{U}$ and $\mathbf{V}$ are correlated. More precisely, for $\mathbf{W} \in \{0, -1, 1\}^{d \times m}$ sampled according to a Gumbel-softmax distribution [43], the entries of $\mathbf{U}$ and $\mathbf{V}$ are constructed from $\mathbf{W}$ as $\mathbf{U}_{ij} = \mathbb{1}\{\mathbf{W}_{ij} = 1\}$ and $\mathbf{V}_{ji} = \mathbb{1}\{\mathbf{W}_{ij} = -1\}$ for $i \in [d], j \in [m]$. Because the entries $\mathbf{U}_{ij}$ and $\mathbf{V}_{ji}$ may never be both equal to 1, there are no self-loops in the induced half-square graph.

**A hybrid likelihood model**  A second important challenge is to propose flexible density models $p_\Theta$ that have reasonable runtime as well as enough capacity for practical purposes. For this, we further exploit the semantics of the factor graph by introducing deterministic factor variables $h_f$ at each factor node $f \in F$. These variables are calculated as the output of a multi-layer perceptron on the input variables of each factor defined by the matrix $\mathbf{U}$, $h_f = \text{MLP}(\mathbf{U}_{:,f} \circ X; \Theta_f)$, for neural networks parameters $\Theta_f$. Then, the conditional distribution of each node depends linearly on its parent factors defined by the matrix $\mathbf{V}$, $X_j \sim \text{Normal}\left(\alpha_j^\top (\mathbf{V}_{:,j} \circ h) + \beta_j, \sigma_j^2\right)$ for parameters $\alpha_j \in \mathbb{R}^m$, $\beta_j \in \mathbb{R}$, $\sigma_j > 0$. The resulting computational gains are highlighted in Figure 3B,C. Although we present here the specific case of a Gaussian likelihood model, the same strategy may be adopted for more complex distributional models such as sigmoidal flows [6].

## 5   Experiments

We tested DCD-FG on both synthetic and real-world data sets with $d = 100$ to $d = 1,000$, and a large number of observations ($n \geq 50,000$). In this large-scale setting, many state-of-the-art causal discovery methods fail to terminate (DCDI and IGSP). Therefore, we compared DCD-FG to NOTEARS [19], its additive non-linear variant NOBEARS [26] and its linear low-rank variant NOTEARS-LR [20]. In order to have a baseline that is external to the NOTEARS framework, we applied IGSP (after feature aggregation via clustering with different resolutions when necessary for a reasonable runtime). For every model, we performed a hyperparameter search using a goodness of fit metric on a small validation set. We provide further details on all experiments, including the grids used for hyperparameter search, as well as supplementary experiments, in Appendix F.

### 5.1   Gaussian Structural Causal Models

We consider synthetic data sets with perfect interventions and known targets. Each data set has $d = 100$ nodes and $n = 50,000$ observations, sampled from interventional distributions governed by either a linear causal mechanism [44] or a nonlinear causal mechanism with additive noise (NN) [45]. Graphs are sampled from an Erdős-Rényi random directed graph model with $m = 10$ factors. A total of $K = 100$ interventions were performed, each sampling up to 3 target nodes. Datapoints from 20 interventional regimes were held-out for model evaluation.

We assessed the performance of each method by the negative log-likelihood of datapoints from unseen interventions (Interventional NLL) [46], as well as two metrics comparing the estimated graph to the ground truth graphs: the structural Hamming distance (SHD), and the precision and recall of edge detection. We report results for 10 randomly generated graphs and data sets in Figure 4. NOTEARS-LR and DCD-FG both outperformed all methods for all metrics on the linear dataset ($p < 0.01$, Wilcoxon signed-rank test), showing that both effectively exploit the low-rank structure

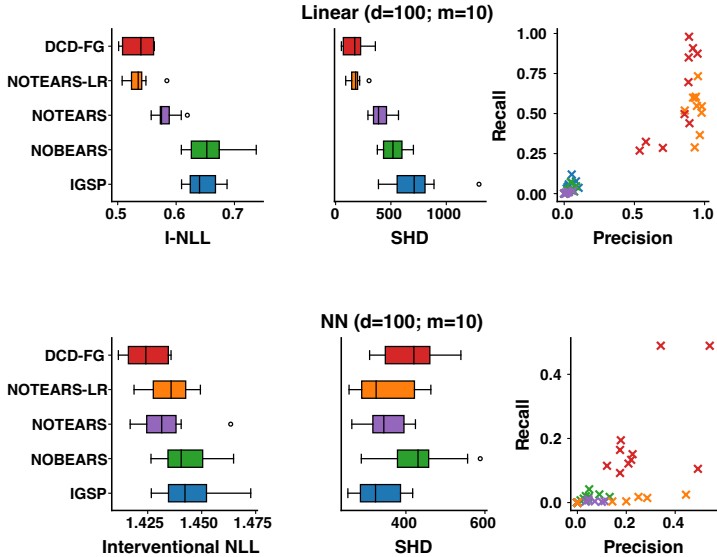

Figure 4: Results on simulated Gaussian structural causal models (Section 5.1).

of the causal graph. On the non-linear (NN) dataset, DCD-FG outperformed all methods in terms of interventional NLL, as well as recall and F1-score (combining precision and recall, Appendix F) ($p < 0.01$, Wilcoxon signed-rank test). DCD-FG has high SHD but we attribute this to the fact that all methods besides DCD-FG discover very sparse graphs.

## 5.2 Genetic Interventions and Gene Expression Data

As an application to real world data, we present an experiment focused on causal learning of gene regulatory networks from gene expression data with genetic interventions, a central problem in modern molecular biology [3, 47, 48]. Although this particular task has been studied by computational biologists for over two decades, there have been substantial experimental advances in the last few years. In particular, a method called Perturb-Seq now allows us to perform interventions targeting hundreds or thousands of genes and measure the effect on full gene expression profiles in hundreds of thousands of single cells using single cell RNA-seq [13]. Surprisingly, however, little to no causal learning work has focused on these advanced datasets. A few notable exceptions, such as [35], focused on early data for which the benchmarked methods were tractable ($d = 24$ genes).

We focus on a recent Perturb-CITE-seq experiment [14] that contains expression profiles from 218,331 melanoma (cancer) cells, after interventions targeting each of 249 genes. Each measurement from a single-cell combines the identity of the intervention (target gene) and a count vector where each entry is the expression level of each gene in the genome. Because of experimental limitations [49], we observe signal only for a subset of several thousand genes (here we selected $d = 1,000$ genes) out of the approximately 20,000 genes in the genome. This dataset includes patient-derived melanoma cells with same genetic interventions but exposed to three conditions: co-culture with T cells derived from the patient's tumor (73,114 cells) (which can recognize and kill melanoma cells), interferon (IFN)-$\gamma$ treatment (87,590 cells) and control (57,627 cells) that we treat as three separate datasets. The goal of the experiment was to identify gene networks in the melanoma cancer cells that either confer resistance or sensitivity to T cell mediated killing, to identify targets for therapeutic intervention in cancer. For every dataset, we retain cells from 20% of the interventions as a test set unavailable during training.

We applied our baseline methods as well as DCD-FG to each of the three datasets. Because we do not have a ground truth causal graph, we use datapoints from held-out interventions to evaluate the models [46], reporting both the interventional NLL (I-NLL) and the mean absolute error (I-MAE) across those interventions (Figure 5). Note that accurately predicting the outcome of genetic interventions that were not measured experimentally is of high utility to biologists.

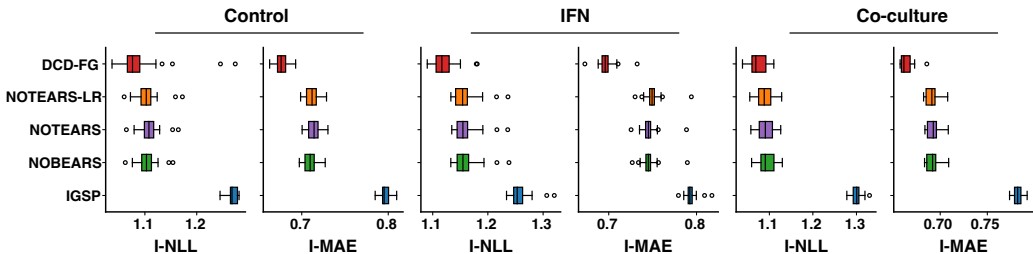

Figure 5: Results on the Perturb-CITE-seq dataset [14] (Section 5.2, lower is better).

DCD-FG outperformed all variants of the NOTEARS method by a large margin, including NOTEARS-LR, and for all metrics ($p < 0.01$, Wilcoxon signed-rank test). In order to diagnose the poor performance of the competing methods, we looked at the number of inferred edges by each method. All variants of NOTEARS identified extremely sparse graphs (less than a hundred edges for NOTEARS and NOBEARS, and a few thousand edges for NOTEARS-LR), which may explain their inability to predict the effect of held-out interventions. Interestingly, IGSP identified hundreds of thousands of edges, a number that was comparable to DCD-FG, but still had poor performance. This suggests that the IGSP-inferred graph did not recapitulate well the causal relationships between genes.

In particular, we carefully examined the $f$-DAG $G_f$ obtained with the best performing model from our hyperparameter sweep on the IFN-$\gamma$ treated cells. That model has $m = 20$ factors, and the half-square $G_f^2[V]$ has 196,303 edges. On average, each module has 194 ingoing edges and 116 outgoing edges. To facilitate visualization, we display the half-square over factors $G_f^2[F]$ in Figure 6.

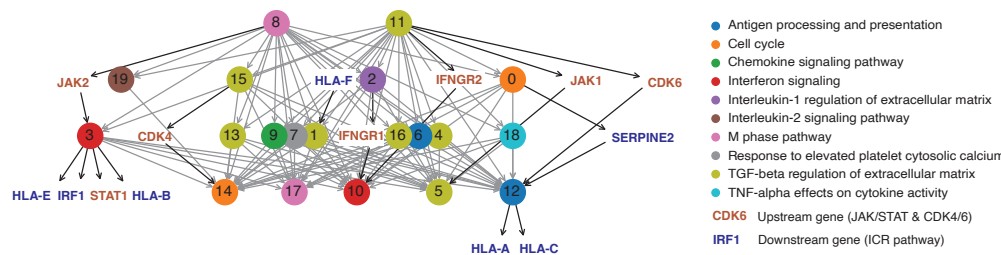

Figure 6: Half-square $G_f^2[F]$ (completed with a few genes) of an $f$-DAG identified by DCD-FG on IFN-$\gamma$ treated malignant cells with interventions. Circles: factors, colored by the gene set enrichment analysis result for its incoming and outgoing genes. Empty nodes: genes, labelled name. Red/blue: genes expected to be up or downstream (resp.) by prior biological knowledge.

To begin to assess the biological relevance of the graph, we performed two analyses. First, we tested the list of incoming and outgoing genes for each factor for enrichment in genes from known biological processes (via [50]). The top hits (node colors in Figure 6) captured many relevant processes in perturbed malignant cells treated by IFN including antigen presentation (needed for recognition by the T cells), multiple innate immune and related signaling pathways (chemokine, interferon, TNF-$\alpha$, and TGF-$\beta$ signaling; all can affect the ability of immune cells to target cancer cells) and stages of the cell cycle. Thus, the graph captured key regulated processes in the response. Second, and more crucially, to highlight key genes in the context of the graph, we displayed them onto the half-square $G_f^2[F]$ based on their strongest link (details in Appendix F). As a proof of concept, we focused on two classes of genes: key known regulators we expect to be positioned upstream, such as the interferon receptors (IFNGR1 and 2) which sense the IFN signal, JAK/STAT, needed to transduce the signal, and CDK4 and 6, which regulate the cell cycle but also repress antigen presentation genes; and others we expect to be downstream, in particular those from an immune cancer resistance pathway we previously discovered in patient tumors (ICR [51]; downstream) in Figure 6. Excitingly, while no such information was used to constrain the model, it captured these ordered relations, including a causal path from the interferon receptors to interferon signaling modules, from JAK to interferon signaling to STAT, IRF1 and HLA genes, from CDK4 to the cell cycle, and from CDK6 and the IFNGR1 to HLA genes (antigen presentation, MHCI genes) ([14, 51] and references therin). Notably,

there are also some connections that may not be borne out biologically, such as the separation of CDK4 / CDK6 to different pathways. Overall, DCD-FG is a promising starting point for deciphering gene regulation at the scale of the whole transcriptome with Perturb-seq data, and predicting the outcome of interventions that were not tested experimentally.

# 6 Discussion

We have proposed DCD-FG, a novel approach for large-scale causal discovery that restricts the search space to factor directed acyclic graphs, and efficiently exploits this structure during inference. Our theoretical results suggest that this class of graphs offers statistical benefits under either the faithfulness assumption or under some stochastic edge perturbation model in random graphs. Our numerical experiments show that in important real-world examples, our method outperforms NOTEARS, NOBEARS as well as low-rank variants of those methods.

Since the publication of NOTEARS [19], two manuscripts highlighted that the method's evaluation may be confounded by the design of the simulations [52, 53]. However, those studies exclusively focus on causal discovery from observational data. The recent results obtained by DCDI on interventional data with a more suitable simulation design [6], as well as the results of this manuscript on real data show the potential promise of the overall framework.

Recently, several other papers identified the acyclicity constraint as a bottleneck for causal discovery learning, and propose to either use the constraint as a soft penalty [54], or discard it from the objective function [55] by using a different parameterization of the causal graph learning approach, and interventional data. Although the approach from [54] is currently restricted to learning linear causal models from observational data, [55] could complement our approach by simplifying the optimization procedure but still explicitly model low-rank interactions.

We presented an application of DCD-FG to a large-scale high-throughput gene expression dataset with genetic perturbations ("Perturb-Seq"). The method had better predictive performance for held-out perturbations than state-of-the-art, and identified both well established relations and new intriguing ones, offering great utility to biologists. Notably, some of the causal relationships may not be accurate, and it is likely that several assumptions of the underlying model may be violated. The biological evaluation and validation of the method will therefore be important to more deeply assess the performance of DCD-FG.

Future work will explore the specification of the noise model, for which count distributions (potentially as part of a latent variable model) may be more appropriate [56], as well as the absence of confounding variables such as cell cycle [14]. Additionally, we plan to investigate extending the framework of DCD-FG to the inference of causal models with feedback loops [57] in order to generate an even more exact and biologically interpretable causal graph. Finally, having a Bayesian alternative of DCD-FG (e.g., based on [58]) would allow scientists to apply those methods for automated experimental design and scientific discovery.

## Acknowledgments and Disclosure of Funding

We thank Sébastien Lachapelle, Philippe Brouillard, Alexandre Drouin, Chandler Squires and Gonçalo Rui Alves Faria for insightful conversations about causal structure learning problems. We thank Natasa Tagasovska, Stephen Ra, and Kyunghyun Cho for general conversations about causal inference and biology. We also acknowledge Katie Geiger-Schuller, Chris Frangieh, Taka Kudo, Josh Weinstock and Basak Eraslan for discussions about Perturb-seq and the Perturb-CITE-seq dataset. We warmly thank Geoffrey Négiar, Natasa Tagasovska, and Tara Chari for their constructive criticisms on draft of this paper.

Disclosures: Romain Lopez and Jan-Christian Huetter are employees of Genentech. Jan-Christian Huetter has equity in Roche. Jonathan Pritchard acknowledges support from grant R01HG008140 from the National Human Genome Research Institute. Aviv Regev is a co-founder and equity holder of Celsius Therapeutics and an equity holder in Immunitas. She was an SAB member of ThermoFisher Scientific, Syros Pharmaceuticals, Neogene Therapeutics, and Asimov until July 31st, 2020; she has been an employee of Genentech since August 1st, 2020, and has equity in Roche.

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
