# Appendices

In Appendix A, we present definitions for $f$-DiGraphs and $f$-DAGs, as well as general properties of these graphs. In particular, we describe the relationship between adjacency matrices of these graphs and low-rank matrices. In Appendix B, we discuss the concept of identifiability of $f$-DAGs and Markov Equivalence Classes (MEC). In particular, we show that for a set of fixed factors $F$ and a growing number of variable nodes, random $f$-DAGs can be identified from observational data. In Appendix C, we prove our main theoretical result on Boolean-rank instability under edge perturbation. In Appendix D, we discuss the time complexity of calculating differentiable acyclicity scores in the case of $f$-DAGs. In Appendix E, we present the implementation details for DCD-FG. In Appendix F, we provide details on our numerical experiments.

## A   Factor Directed Acyclic Graphs

In this section, we present general definitions and properties of Factor Directed Graphs.

### A.1   Definitions

Let $d \in \mathbb{N}$ be the number of feature vertices and $m \in \mathbb{N}$ be the number of factor vertices. We start with a few definitions.

**Definition 1.** *(Factor Directed Graph) Let $V = \{v_1, \ldots, v_d\}$ denote the set of* feature vertices *(which we also call* feature nodes, variable vertices, *or* variable nodes*), and $F = \{f_1, \ldots, f_m\}$ the set of* factor vertices *(or* factor nodes*). Let $E$ be a set of directed edges, such that there are no edges between two nodes of same type (factors or features). We define a* Factor Directed Graph *($f$-DiGraph) as the bipartite graph $G_f = (V, F, E)$.*

**Definition 2.** *(Set of factor graphs with $m$ modules and $d$ nodes) The subset of factor graphs $\mathbb{G}_d^m$ with $d$ variables and $m$ factors is defined as:*

$$\mathbb{G}_d^m = \{G_f \mid G_f = (V, F, E) \text{ is a factor directed graph and } |V| = d, |F| = m\}. \tag{9}$$

**Definition 3.** *(Half-square graphs and induced feature graph) $G_f$ canonically induces two half-square graphs $G_f^2[V]$ and $G_f^2[F]$. $G_f^2[V]$ is defined as the graph on $V$ with an edge between two vertices of $V$ if there is a path of length exactly two between these vertices in $G_f$. $G_f^2[F]$ is defined similarly on the nodes $F$. We specifically write $G = G_f^2[V]$ and refer to it as the feature graph (as opposed to the factor graph $G_f$).*

**Definition 4.** *(Set of feature graphs with $m$ modules and $d$ nodes) The set of feature graphs $\mathcal{G}_d^m$ formed from half-square graphs with $d$ variables and $m$ factors is defined as:*

$$\mathcal{G}_d^m = \{G = G_f^2[V] \mid G_f \in \mathcal{G}_d^m\}. \tag{10}$$

### A.2   General properties

We start by outlining some general properties of $f$-DiGraphs. First, we observe that every graph may be written as the feature graph of a bipartite graph.

**Proposition 2.** *(Representation) Let $\mathcal{G}$ denote the set of all feature graphs, i.e., directed graphs on $V$. Then we have*

$$\mathcal{G} = \bigcup_{m=1}^{\infty} \mathcal{G}_d^m. \tag{11}$$

*Proof.* By definition, we have that $\mathcal{G}_d^m \subset \mathcal{G}$ for all $m$, which proves the reverse set inclusion. Therefore, we focus on the direct set inclusion. Let $G = (V, E) \in \mathcal{G}$. We set $F = E$ and define the set of edges $E'$ as

$$E' = \bigcup_{e=(v_1, v_2) \in E} \{(v_1, e), (e, v_2)\}. \tag{12}$$

Then, for the bipartite graph $G_f' = (V, F, E') \in \mathbb{G}_d^{|E|}$, we obtain $G_f'^2[V] = G$ as desired. $\square$

Second, we discuss the size of the set of DAGs within $\mathcal{G}_d^m$ compared to $\mathcal{G}$. We start with a lemma for a fixed topological ordering $\sigma$.

**Lemma 2.** *Denote by $\mathcal{D}_d$ the subset of acyclic graphs in $\mathcal{G}$ with $d$ nodes, and by $\mathcal{D}_d^m$ the subset of acyclic graphs in $\mathcal{G}_d^m$. For $\sigma$, a permutation of $\{v_1, \ldots, v_d\}$, we denote by $\mathcal{D}_d^m(\sigma)$ (resp. $\mathcal{D}_d(\sigma)$) the subset of graphs in $\mathcal{D}_d^m$ (resp. $\mathcal{D}_d$) for which $\sigma$ is a topological ordering. Then, we have:*

$$|\mathcal{D}_d(\sigma)| = 2^{\frac{d(d-1)}{2}}, \tag{13}$$

$$|\mathcal{D}_d^m(\sigma)| \le \binom{d+m}{m} 2^{dm}. \tag{14}$$

*Proof.* If $\sigma$ is a valid topological ordering for a graph, then its adjacency matrix is upper triangular (with zero on the diagonal) under this ordering of the rows and columns.

In the case of $\mathcal{D}_d(\sigma)$, we must simply count the number of binary matrices that are upper triangular with zeros on the diagonal. As such matrices have $\frac{d(d-1)}{2}$ entries potentially taking one of two values, this proves the first part of this proposition.

The case for $\mathcal{D}_d^m(\sigma)$ is slightly more technical. In this setting, we have $d$ variable nodes, and $m$ factors. We are able to arrange the factors in between the features to obtain all possible topological orderings of factor graphs compatible with $\sigma$,

$$(v_1, \ldots, v_{i_1}, f_1, v_{i_1+1}, \ldots, v_{i_m}, f_m, v_{i_m+1}, \ldots, v_d), \tag{15}$$

with $1 \le i_1 < \ldots < i_m \le d$, where, without loss of generality, we assumed the identity permutation on factors and features separately. We note that there are $\binom{d+m}{m}$ such possible combinations.

For a fixed arrangement, we may now observe that a variable node $v_a$ can only be connected to one or several factor nodes appearing before $v_a$ in the topological ordering, and each factor node $f_j$ may only be connected to one or several feature nodes appearing before $f_j$ in the topological ordering. We define the number of such possible adjacency patterns as $I$.

More precisely, each factor vertex $f_j$ has $2^{i_j}$ potential incoming edge patterns. Also, each feature vertex $v_a$ appearing after $f_j$ gives rise to $2^j$ potential edge patterns and there are $i_{j+1} - i_j$ such vertices. Therefore, writing $i_{m+1} = d$, we have:

$$\log_2 I = \sum_{j=1}^m i_j + \sum_{j=1}^m j(i_{j+1} - i_j) \tag{16}$$

$$= \sum_{j=1}^m i_j + \sum_{j=1}^{m-1} j i_{j+1} + dm - \sum_{j=1}^m j i_j \tag{17}$$

$$= dm + \sum_{j=1}^m i_j + \sum_{j=2}^m (j-1) i_j - \sum_{j=2}^m j i_j - i_1 \tag{18}$$

$$= dm + \sum_{j=1}^m i_j - \sum_{j=1}^m i_j + \sum_{j=2}^m j i_j - \sum_{j=2}^m j i_j \tag{19}$$

$$= dm. \tag{20}$$

Because several factor graphs may yield the same half-square graph, we only have an upper bound, as claimed,

$$|\mathcal{D}_d^m(\sigma)| \le \binom{d+m}{m} 2^{dm}. \qquad \square$$

We can use this lemma to prove the following bound for the cardinality of the entire set of graphs:

**Proposition 3.** *(Cardinality of Directed Acyclic Graphs and half-square of f-DAGs) Denote by $\mathcal{D}_d$ the subset of acyclic graphs in $\mathcal{G}$ with $d$ nodes, and by $\mathcal{D}_d^m$ the subset of acyclic graphs in $\mathcal{G}_d^m$. For $d \ge m$, we have the inequality*

$$\frac{|\mathcal{D}_d^m|}{|\mathcal{D}_d|} \le \left(1 + \frac{d}{m}\right)^{m+1/2} \exp\left\{-\frac{d^2}{2} + d\left(m\log 2 + \log(m+d) + \frac{\log 2}{2} - 1\right)\right\} \tag{21}$$

$$\le \exp\left\{dm + 4d\log(m+d) - \frac{\log 2}{2}d^2\right\}, \tag{22}$$

*and therefore for fixed $m$, $\frac{|\mathcal{D}_d^m|}{|\mathcal{D}_d|} \to 0$ when $d \to +\infty$.*

*Proof.* Let us first provide a (loose but sufficient) lower bound on $|\mathcal{D}_d|$. In the light of the previous result for a fixed permutation, there are at last $2^{\frac{d(d-1)}{2}}$ DAGs in $|\mathcal{D}_d|$.

Next, we want to establish an upper bound on $|\mathcal{D}_d^m|$. Because for each permutation $\sigma$, there are at most $|\mathcal{D}_d^m(\sigma)|$ distinct graphs, an upper bound on $|\mathcal{D}_d^m|$ is given by $d!\binom{d+m}{m}2^{dm}$.

Putting this together, and using a non-asymptotic version of Stirling's formula, we obtain

$$\frac{|\mathcal{D}_d^m|}{|\mathcal{D}_d|} \leq \frac{(d+m)!}{m!}2^{dm}2^{-\frac{d(d-1)}{2}} \tag{23}$$

$$\leq \frac{\sqrt{2\pi(m+d)}(m+d/e)^{m+d}}{\sqrt{2\pi m}(m/e)^m}2^{dm}2^{-\frac{d(d-1)}{2}}e^{\frac{1}{12(m+d)}-\frac{1}{12m+1}} \tag{24}$$

$$\leq (m+d)^{m+d+\frac{1}{2}}2^{dm-\frac{d(d-1)}{2}} \tag{25}$$

$$\leq \exp\left\{\left(m+d+\frac{1}{2}\right)\log(m+d) + dm\log 2 - \frac{\log 2}{2}d^2 + \frac{d\log 2}{2}\right\} \tag{26}$$

$$\leq \exp\left\{dm + 4d\log(m+d) - \frac{\log 2}{2}d^2\right\} \tag{27}$$

if $d \geq m$. By comparing coefficients inside the exponential, for fixed $m$, this upper bound tends to $0$ as $d \to \infty$. $\qquad\square$

### A.3  Relationship to low-rank matrices

We now discuss properties of the mapping $\zeta : G_f \mapsto G = G_f^2[V]$ defined over the set of factor graphs.

First, we remark that, in general, $\zeta$ is not injective, as shown by the following counterexample.

**Example 1.** *(Induced half-square is not injective) Let $V = \{v_1, v_2, v_3\}$ and $F = \{f_1, f_2\}$. Let $E_1 = \{(v_3, f_2), (v_3, f_1), (v_2, f_2)\} \cup \{(f_1, v_2), (f_2, v_1)\}$ and $E_2 = \{(v_3, f_2), (v_2, f_1)\} \cup \{(f_1, v_1), (f_2, v_1), (f_2, v_2)\}$. For $G_{f,1} = (V, F, E_1)$ and $G_{f,2} = (V, F, E_2)$, we have the identity $G_{f,1}^2[V] = G_{f,2}^2[V]$, in the sense that both graphs have the same set of vertices and edges.*

Second, we characterize the image $\zeta(\mathbb{G}_d^m) = \mathcal{G}_d^m$ in terms of the Boolean rank of the associated adjacency matrices. Let us define the adjacency matrix $\mathcal{A}(G_f)$ of a factor graph $G_f \in \mathbb{G}_d^m$. Because the graph is bipartite, up to a permutation of the rows and columns, we may write it in block form,

$$\mathcal{A}(G_f) = \begin{bmatrix} \mathbf{0}_{d\times d} & \mathbf{U} \\ \mathbf{V} & \mathbf{0}_{m\times m} \end{bmatrix}, \tag{28}$$

where $\mathbf{U} \in \mathbb{R}^{d\times m}$ (resp. $\mathbf{V} \in \mathbb{R}^{m\times d}$) denotes the binary matrix encoding the presence or absence of edges towards factor nodes (resp. variable nodes) according to the edge set $E$. We now relate these two matrices to the adjacency matrix of the half-square graph.

**Proposition 1** (Bounded Boolean rank of half-square adjacency matrix). *For a factor graph $G_f = (V, F, E)$, let $\mathbf{U} \in \{0,1\}^{d\times m}$ (resp. $\mathbf{V} \in \{0,1\}^{m\times d}$) be the binary matrix encoding the presence or absence of edges directed towards factor nodes (resp. variable nodes), according to $E$. The adjacency matrix $\mathcal{A}(G)$ of the half-square graph $G$ may be decomposed as $\mathcal{A}(G) = \mathbf{U} \diamond \mathbf{V}$ where $\diamond$ denotes the Boolean matrix product, $(\mathbf{U} \diamond \mathbf{V})_{ij} = \bigvee_{k=1}^m \mathbf{U}_{ik} \wedge \mathbf{V}_{kj}$, $i, j \in [d]$. Consequently, $\mathcal{A}(G)$ has Boolean rank bounded above by $m$. Conversely, every adjacency matrix over the feature nodes with Boolean rank bounded by $m$ can be written as half-square of an $f$-DiGraph with $m$ factors.*

*Proof.* Let $G_f$ be a factor graph, and let $\mathbf{U}$ and $\mathbf{V}$ be defined as in (28). The adjacency matrix $\mathcal{A}(G)$ of the half-square graph $G = G_f^2[V]$ can be calculated as

$$\forall (i,j) \in [d]^2, \quad \mathcal{A}(G)_{ij} = \bigvee_{k=1}^m \mathbf{U}_{ik} \wedge \mathbf{V}_{kj}, \tag{29}$$

where $\wedge$ and $\vee$ denote the logical AND and OR operators, respectively.

Indeed, by definition, there is an edge between two nodes $v_a$ and $v_b$ of $G$ if and only if there exists a path of length two between those two nodes in $G_f$. Because edges may exist only between factor and feature nodes, this condition is met if and only if there exists an edge between $v_a$ and $f$ as well as $f$ and $v_b$ for at least one factor node $f$.

This proves that every adjacency matrix of a graph in $\mathcal{G}_d^m$ can be written as the matrix product of a $d \times m$ and a $m \times d$ matrix for the Boolean arithmetic. By definition [29], $\mathcal{A}(G)$ therefore has Boolean rank bounded above by $m$.

The converse follows by observing that, by definition, an adjacency matrix with Boolean rank bounded above by $m$ admits the decomposition (29) with binary matrices $\mathbf{U}$, $\mathbf{V}$. Defining an $f$-DiGraph with these adjacency matrices according to (28) yields the claim. $\square$

For one direction of Proposition 1, we have a similar result for the weighted adjacency matrices $\mathbf{UV}$.

**Proposition 4** (Bounded rank of weighted half-square adjacency matrix). *For a factor graph $G_f$ and matrices $\mathbf{U}$ and $\mathbf{V}$, the (regular matrix) product $\mathbf{UV}$ counts the number of paths of length two between two variable nodes in $G_f$, and therefore is a valid (weighted) adjacency matrix for $G = G_f^2[V]$. Additionally, $\mathbf{UV}$ has matrix rank bounded above by $\min(d, m)$.*

*Proof.* Replacing $\vee$ by $+$ and $\wedge$ by $\times$ in (29), we obtain the matrix product $\mathbf{UV}$. Because of this, individual entries in the matrix $\mathbf{UV}$ indeed count the number of distinct paths of length two between two feature nodes in $G_f$. As a consequence, $\mathbf{UV}$ has a zero entry if and only if $\mathcal{A}(G)$ has a zero entry. This proves that $\mathbf{UV}$ is a valid weighted adjacency matrix. By construction, its matrix rank is bounded above by $\min(d, m)$. $\square$

### A.4 Alternate definitions of low-rank graphs

Here, we introduce various notions of low-rank constraints on graphs, and make precise the class we are concerned with in this paper. Towards this end, one could consider a variety of subsets of $\mathcal{G}$, the set of all graphs on $d$ nodes, in particular:

1. $\mathcal{G}_{\text{lin}}^m$: graphs that admit a weighted adjacency matrix $W$ that has a matrix factorization of rank $\leq m$.

2. $\mathcal{G}_{\text{lin,nonneg}}^m$: graphs that admit a weighted adjacency matrix $W$ that has a non-negative matrix factorization of rank $\leq m$.

3. $\mathcal{G}_{\text{bool}}^m$: graphs that admit a weighted adjacency matrix $W$ that has a Boolean matrix factorization of rank $\leq m$.

Note that in the definition of $\mathcal{G}_{\text{lin}}^m$, we allow for $W$ to encode the presence of an edge in $G$ with any non-zero entry, positive or negative. In this context, we have the following set inclusions:

$$\mathcal{G}_{\text{bool}}^m = \mathcal{G}_{\text{lin,nonneg}}^m \subsetneqq \mathcal{G}_{\text{lin}}^m \subsetneqq \mathcal{G}, \tag{30}$$

for $m < d$. To understand this result, it is important to note that for a given matrix, its non-negative rank and Boolean rank do not necessarily coincide, but $\mathcal{G}_{\text{bool}}^m = \mathcal{G}_{\text{lin,nonneg}}^m$ since we allow for arbitrary weighted adjacency matrices in the definition of $\mathcal{G}_{\text{lin,nonneg}}^m$.

The previous low-rank work [20] searches for graphs in $\mathcal{G}_{\text{lin}}^m$. By contrast, we do not consider $\mathcal{G}_{\text{lin}}^m$, but instead choose to exclusively work with $\mathcal{G}_{\text{bool}}^m$. Considering $\mathcal{G}_{\text{bool}}^m$ instead of $\mathcal{G}_{\text{lin}}^m$ gives further rise to an intuitive way of restricting the nonlinear functional relationships on top of the graphical structure while maintaining low asymptotic computational complexity, as presented in Section 4. Besides being more immediate when starting from a linear structural equation model, we see no particular reason for favoring $\mathcal{G}_{\text{lin}}^m$ over $\mathcal{G}_{\text{bool}}^m$ in light of the practical benefits outlined in the later sections of this paper. Moreover, the only linear low-rank models not captured in $\mathcal{G}_{\text{bool}}^m$ are those in which the contributions of multiple factors cancel out to produce more zeros than expected from the sparsity pattern of the factors $U, V$, corresponding to a lack of faithfulness of the factor graph. We consider these models edge cases that could safely be excluded from the search space.

# B  Identifiability of $f$-DAGs

In this section, we investigate the identifiability of Factor Directed Acyclic Graphs from observational data in the context of causal discovery. We first define the concept of Markov Equivalence Class (MEC), and we introduce a Boolean-rank restricted equivalence class. We show that the latter is in general smaller (and sometimes strictly smaller) than the MEC.

## B.1  Markov equivalence classes

Under the classical set of assumptions commonly employed in causal discovery, namely causal sufficiency, causal Markov property, and faithfulness, we may identify the causal DAG from observational data only up to its Markov Equivalence Class (MEC) [59]:

**Definition 5.** *(Markov Equivalence Class) The MEC of the half-square graph $D \in \mathcal{D}_d^m$ is $M(D) = \{D' \mid D' \sim D\}$ where $\sim$ denotes Markov equivalence.*

A graphical rule for Markov equivalence is that two graphs are Markov equivalent if they share the same skeleton and v-structures [59]. Ideally, we wish to characterize the complexity of searching for a DAG only in a subset of the Markov equivalence class that is composed of graphs in $\mathcal{D}_d^m$.

**Definition 6.** *($f$-MEC) The $f$-MEC $M_f^m(D)$ of a half-square graph $D \in \mathcal{D}_d^m$ is the set of DAGs that are Markov equivalent to $D$ and arise as the half-square of a factor graph with at most $m$ factors: $M_f^m(D) = M(D) \cap \mathcal{D}_d^m$.*

We represent a MEC with an essential graph, defined as the union of all vertices and edges in the MEC. In particular, we say that an edge is unoriented if there exists an edge and its reverse orientation in the MEC.

Although by definition, we have the inclusion $M_f^m(D) \subset M(D)$, it is worth noting that this inclusion is in general not equal.

**Example 2.** *Let $V = \{v_1, v_2, v_3\}$, $F = f_1$ and $E = \{(v_3, f_1), (f_1, v_1), (f_1, v_2)\}$. The graph $G_f = (V, F, E)$ is a valid factor graph with one factor. The half-square graph $G$ has two edges $\{(v_3, v_1), (v_3, v_2)\}$ that are both unoriented for the (classical) Markov equivalence class. Indeed, $M(G)$ contains three graphs, obtained by flipping edges (except for the configuration that creates a v-structure). However, $M_f^1(G)$ has only a unique graph, because the two other graphs in $M(G)$ require two factors when being described as a factor graph.*

We leave the problem of providing an algebraic characterization of an $f$-MEC as future work.

## B.2  Identifiability of Boolean low-rank graphs

Because graphs in $\mathcal{D}_d^m$ potentially contain many v-structures, we obtain a simple condition for identifiability.

**Lemma 3.** *(Unoriented edges in Boolean low-rank graphs) Let $D_f = (V, F, E)$ be an $f$-DAG and $D = D_f^2[V] \in \mathcal{D}_d^m$. For every unoriented edge $(v_i, v_j)$ in the essential graph of $D$, there exists a factor $f$ with unique parent $v_i$. Consequently, if every factor $f \in F$ has at least two parents in $D_f$, then the MEC of $D$ reduces to the singleton $\{D\}$.*

*Proof.* Let $D_f$ be a factor directed graph such that $D = D_f^2[V]$. Let $(v_i, v_j)$ be an edge of $D_f$ that is unoriented in $M(D)$. By definition of the factor directed graph, there exists $f \in \mathcal{F}$ such that the edges $(v_i, f)$ and $(f, v_j)$ are part of $\mathcal{G}$. Because the edge is unoriented, it cannot be part of a v-structure. Consequently, there are no other feature vertices $v_k$ connected to $f$ or it would create a v-structure in $D$.

In the case that every factor $f \in \mathcal{F}$ has at least two parents, there can be no unoriented edges, and therefore the graph is identifiable, i.e., it is alone in its MEC. $\qquad\square$

In order to quantify how frequent this configuration is, we introduce a random factor directed acyclic graph model, inspired by the Erdős-Rényi model [36]:

**Definition 7.** *(Random sequence of growing $f$-DAGs) Let $(D_{f,d})_{d=0}^{\infty}$ denote a random sequence of factor directed graphs defined recursively, with $D_{f,0} = (\varnothing, F, \varnothing)$ and $|F| = m$. Let $D_{f,d} = (V_d, F, E_d)$ be the graph at step $d$, and $\sigma_d$ a permutation of $F \cup V_d$ that specifies a topological ordering $\{\sigma(v_1), \ldots, \sigma(v_d)\}$. We define $V_{d+1} = V_d \cup \{v_{d+1}\}$ where $v_{d+1}$ denotes a new variable node. We extend the permutation $\sigma_d$ into a new permutation $\sigma_{d+1}$ of $F \cup V_{d+1}$ by randomly inserting the node $v_{d+1}$ into the linear order induced by $\sigma_d$ (out of the $d + m + 1$ possible choices). To obtain $E_{d+1}$, we add to $E_d$ edges of the form $(v_{d+1}, f)$ (resp. $(f, v_{d+1})$) if $(\sigma(v_{d+1}) \leq \sigma(f))$ (resp. $(\sigma(f) \leq \sigma(v_{d+1}))$) independently with probability $p \in (0, 1)$. Finally, we define $D_{f,d+1} = (V_{d+1}, F, E_{d+1})$ and have $D_{d+1} = D_{f,d+1}^2[V] \in \mathcal{D}_{d+1}^m$.*

We show that the probability of having at least one unoriented edge in $M(D_d)$ for a fixed $m$ is small for large $d$.

**Proposition 5.** *For $D_d$ sampled according to Definition 7, the size of the MEC (and, by inclusion, of the $f$-MEC) converges to 1 with high probability for fixed $m$:*

$$\mathbb{P}(|M(D_d)| = 1) \geq 1 - \phi(d, m, p), \tag{31}$$

*with $\phi(d, m, p) \to 1$ when $p$ and $m$ are fixed, and $d \to \infty$. Consequently, as the number of variable nodes $d$ grows, the DAG becomes identifiable.*

*Proof.* According to the graphical rule in Lemma 3, if all factors $f \in F$ have at least two parents then the MEC $M(D_n)$ of the DAG $D_n$ reduces to a singleton $\{D_n\}$. Consequently, we have the lower bound

$$\mathbb{P}(|M(D_d)| = 1) \geq \mathbb{P}(\forall f \in F : |\mathrm{Pa}(f)| \geq 2). \tag{32}$$

Calculating the right-hand side of the previous inequality essentially corresponds to calculating the distribution of the number of parents $|\mathrm{Pa}(f)|$ for each factor $f$. To bound this probability, we first note that we can obtain the probability distribution (on graphs) in Definition 7 equivalently by fixing an order on the feature nodes and inserting $m$ factor nodes randomly, one by one, into the order, finally assigning edges independently as before. In this construction, for each factor $f$, the number of preceding feature nodes and the number of parents are independent. The number of parents follows a binomial distribution with parameters $(K_f, p)$ where $K_f$ is the number of feature nodes $K_f$ appearing before $f$ in the topological ordering $\sigma_d$. By construction, $K_f$ is independently sampled from a uniform distribution $K_f \sim \mathrm{Categorical}(\{0, \ldots, d\})$. Therefore, it follows that

$$\mathbb{P}(\forall f \in F : |\mathrm{Pa}(f)| \geq 2) = [\mathbb{P}(\text{a fixed } f \in F \text{ verifies } |\mathrm{Pa}(f)| \geq 2)]^m \tag{33}$$

$$= [1 - \mathbb{P}(\text{a fixed } f \in F \text{ verifies } |\mathrm{Pa}(f)| \in \{0, 1\})]^m \tag{34}$$

$$\geq 1 - m\mathbb{P}(A_d), \tag{35}$$

where we introduce the probabilistic event $A_d = \{\text{a fixed } f \in F \text{ verifies } |\mathrm{Pa}(f)| \in \{0, 1\}\}$. Then, conditioning on $K_f$, we obtain

$$\mathbb{P}(A_d) = \frac{1}{d+1} \sum_{k=0}^{d} \left[ (1-p)^k + kp(1-p)^{k-1} \right] \tag{36}$$

$$= \frac{1}{d+1} \sum_{k=0}^{d} (1-p)^k + \frac{p}{(d+1)(1-p)} \sum_{k=0}^{d} k(1-p)^k \tag{37}$$

$$= \frac{1 - (1-p)^{d+1}}{(d+1)p} + \frac{p}{(d+1)(1-p)} \left[ \frac{1-p}{p^2}(1 - (1-p)^{d+1}) - \frac{d+1}{p}(1-p)^{d+1} \right] \tag{38}$$

$$= \frac{2(1 - (1-p)^{d+1})}{(d+1)p} - (1-p)^d, \tag{39}$$

where we recognized the closed-form expression of a geometric series, and a finite arithmetico-geometric series, respectively. Hence, because $\mathbb{P}(A_d) \to 1$ when $d \to \infty$, we have that $\mathbb{P}(|M(D_d)| = 1) \to 1$ when $d \to \infty$. $\qquad\square$

### B.3 Empirical validation

To simulate $f$-DAGs, we used a random graph model equivalent to Definition 7 and specified as follows. For $d$ variable nodes and $m$ factor nodes, we sample a random permutation $\sigma$ that specifies a

topological ordering on $[d+m]$. For each node $i$ in this topological ordering $\{\sigma(1),\ldots,\sigma(d+m)\}$, we draw the absence or the presence of an edge between node $i$ and each of its potential parents based on a Bernoulli distribution. If node $i$ is a variable node, then it may only be connected to factors present in $\{\sigma(1),\ldots,\sigma(i)\}$ with probability $p_v$. Similarly, if node $i$ is a variable node, then it may only be connected to factors present in $\{\sigma(1),\ldots,\sigma(i)\}$ with probability $p_f$. This is a natural extension of the simulations provided in [6], further adding the factor semantic.

Using this framework, we sampled $T=50$ $f$-DAGs with $d$ nodes and $m$ modules ($p_v=p_f=0.5$), and calculated the size of the MEC using the causaldag Python package (Figure 2A).

## C   Boolean-rank instability under edge perturbation

In this section, we prove the Boolean-rank instability result (Theorem 1). The key idea behind the proof is to identify a sufficient condition for the perturbation to induce a Boolean rank increase in the half-square graph that happens often for large $d$ and fixed $m$. The proof consists of four parts. First, we introduce a technical lemma regarding the overlap of random sets. Namely, any pair of distinct union of sets of random subsets from an alphabet differ in at least two elements with high probability (Lemma 4). Second, we introduce another lemma showing that the sampling of patterns in a binary vector becomes exhaustive after enough independent draws (Lemma 5). Third, we relate those two conditions to a sufficient condition for the increase of the Boolean rank of the half-square of a $f$-DiGraph (Lemma 6). Putting everything together, we finally prove the main theorem (Theorem 1).

### C.1   Random subset non-overlapping lemma

**Lemma 4.** *Let $\{\mathcal{U}_1,\ldots,\mathcal{U}_m\}$ be $m$ random subsets of the alphabet $\Omega=[d]$, where the inclusion of each letter $i\in\Omega$ in each subset $\mathcal{U}_k\subset\Omega$ for $k\in[m]$ is defined by sampling a Bernoulli random variable with parameter $p$, independently over $k\in[m]$ and $i\in[d]$. Further, let $S_1\neq S_2$ be two distinct fixed subsets of $[m]$. Then, the two sets defined as the unions of subsets of $\{\mathcal{U}_1,\ldots,\mathcal{U}_m\}$ indexed by $S_1$ and $S_2$ have low probability of completely overlapping:*

$$\mathbb{P}\left(\left|\left(\bigcup_{j\in S_1}\mathcal{U}_j\right)\triangle\left(\bigcup_{j'\in S_2}\mathcal{U}_{j'}\right)\right|\leq 1\right)\leq\gamma q_A^d, \tag{40}$$

*where $\triangle$ denotes the symmetric set difference, $\gamma=1+{}^d\!/_{(1-p(1-p)^m)}>0$, and $q_A=1-p(1-p)^m\in(0,1)$.*

*Proof.* We introduce $X_w^{S_i}$, the random variable that denotes whether letter $w\in\Omega$ is present in $\bigcup_{j\in S_i}\mathcal{U}_j$. We observe that a letter $w$ gives rise to an element in the symmetric difference of the two sets $S_1$ and $S_2$ if $X_w^{S_1}\neq X_w^{S_2}$. Although the sets $\{\mathcal{U}_1,\ldots,\mathcal{U}_m\}$ are constructed independently, there may be correlations in the union sets we consider in the case of overlap between $S_1$ and $S_2$. Therefore, we decompose the sets $S_1$ and $S_2$ into their overlapping and non-overlapping parts:

$$S_1=S_1'\sqcup S_3 \tag{41}$$
$$S_2=S_2'\sqcup S_3, \tag{42}$$

where $\sqcup$ denotes a disjoint union, $S_3=S_1\cap S_2$, $S_1'=S_1\smallsetminus S_3$ and $S_2'=S_2\smallsetminus S_3$. Under these conditions, we may write

$$A_w:=\{X_w^{S_1}\neq X_w^{S_2}\}=\underbrace{\left\{w\in\bigcup_{j\in S_1}\mathcal{U}_j\cap w\notin\bigcup_{j\in S_2}\mathcal{U}_j\right\}}_{=:\bar{A}_w}\cup\left\{w\in\bigcup_{j\in S_2}\mathcal{U}_j\cap w\notin\bigcup_{j\in S_1}\mathcal{U}_j\right\}. \tag{43}$$

Noticing that these events are symmetric in $S_1$ and $S_2$, we focus on the first one which we denote by $\bar{A}_w$. In each of the above events, $w$ cannot be in the union over $S_3$ because in that case, it would be in the union over both $S_1$ and $S_2$. Therefore, we can decompose $\bar{A}_w$ as

$$\bar{A}_w=\{w\in\bigcup_{j\in S_1'}\mathcal{U}_j\}\cap\{w\notin\bigcup_{j\in S_2'}\mathcal{U}_j\}\cap\{w\notin\bigcup_{j\in S_3}\mathcal{U}_j\}. \tag{44}$$

By the independence of $\{\mathcal{U}_1, \ldots, \mathcal{U}_m\}$, we have that

$$\mathbb{P}(\bar{A}_w) = \left(1 - (1-p)^{|S_1'|}\right)(1-p)^{|S_2'|}(1-p)^{|S_3|} \tag{45}$$

$$= \left(1 - (1-p)^{|S_1'|}\right)(1-p)^{|S_2|}. \tag{46}$$

Then, because $|S_1'|, |S_2| \in \{0, \ldots, m\}$, we have the bound

$$\mathbb{P}(A_w) \geq \mathbb{P}(\bar{A}_w) \geq p(1-p)^m. \tag{47}$$

Now, noting the independence of the letters, we have that

$$\mathbb{P}\left(\left\|\left(\bigcup_{j \in S_1} \mathcal{U}_j\right) \Delta \left(\bigcup_{j' \in S_2} \mathcal{U}_{j'}\right)\right\| \leq 1\right) = \mathbb{P}(\text{At most one } A_w \text{ is true}) \tag{48}$$

$$= (1 - \mathbb{P}(A_w))^d + d\mathbb{P}(A_w)(1 - \mathbb{P}(A_w))^{d-1} \tag{49}$$

$$\leq (1 - \mathbb{P}(A_w))^d + d(1 - \mathbb{P}(A_w))^{d-1} \tag{50}$$

$$\leq (1 - p(1-p)^m)^d + d(1 - p(1-p)^m)^{d-1} \tag{51}$$

$$\leq \left(1 + \frac{d}{1 - p(1-p)^m}\right)(1 - p(1-p)^m)^d. \tag{52}$$

This concludes the proof. $\qquad\square$

## C.2 Exhaustive binary pattern coverage lemma

**Lemma 5.** *Let $\mathcal{V} = [v_1, \ldots, v_m] \in \{0,1\}^m$ be a random vector with each entry independently sampled from a Bernoulli distribution with parameter $p$. Denote by $B$ the following probabilistic event:*

$$B = \{\exists x \in \{0,1\}^m \text{ that is not observed at least twice in } d \text{ independent draws from } \mathcal{V}\}.$$

*Then, the probability of $B$ is small for large $d$, in particular,*

$$\mathbb{P}(B) \leq \delta q_B^d \tag{53}$$

*with $\delta = 2^m \left(1 + \frac{d}{1 - \max\{p^m, (1-p)^m\}}\right)$ and $q_B = 1 - \min\{p^m, (1-p)^m\} \in (0,1)$.*

*Proof.* By a union bound, we obtain

$$\mathbb{P}(B) = \mathbb{P}\left(\exists x \in \{0,1\}^m : x \text{ appears at most once in } d \text{ draws}\right) \tag{54}$$

$$\leq 2^m \max_{x \in \{0,1\}^m} \mathbb{P}\left(\text{a fixed } x \in 2^m \text{ appears at most once in } d \text{ draws}\right). \tag{55}$$

We simply need to upper bound this probability. For a fixed $x \in \{0,1\}^m$, denote by $|x|$ the number of non-zero entries in the binary vector $x$. We then have, by independence:

$$\mathbb{P}\left(\text{a fixed } x \in 2^m \text{ appears at most once in } d \text{ draws}\right) = (1 - p_x)^d + dp_x(1 - p_x)^{d-1}, \tag{56}$$

where

$$p_x = p^{|x|}(1-p)^{m-|x|} \in (p_\beta, p_\alpha), \tag{57}$$

with $p_\alpha = \max\{p^m, (1-p)^m\}$ and $p_\beta = \min\{p^m, (1-p)^m\}$. This inequality directly follows from distinguishing the cases $p \leq 1/2$ and $p > 1/2$. Plugging this into the first bound above, we have

$$\mathbb{P}(B) \leq 2^m \left(1 + \frac{d}{1 - p_\alpha}\right)(1 - p_\beta)^d, \tag{58}$$

which concludes the proof. $\qquad\square$

## C.3 Boolean rank increase lemma

**Lemma 6.** *Let $G_f \in \mathcal{G}_d^m$ be a (fixed) $f$-DiGraph, with partial adjacency matrices $\mathbf{U}$ and $\mathbf{V}$. We denote by $G$ its half-square $G = G_f^2[V]$. Let us state two assumptions.*

*First, we say that the matrix $\mathbf{U}$ satisfies the non-overlapping condition if*

$$\forall (x_1, x_2) \in \{0,1\}^m, x_1 \neq x_2 \implies d_{\mathcal{H}}(\mathbf{U} \diamond x_1, \mathbf{U} \diamond x_2) \geq 2 \qquad \text{(non-overlap)}$$

*where $d_{\mathcal{H}}$ denotes the Hamming distance between two binary vectors.*

*Second, we say that the matrix $\mathbf{V}$ satisfies the coverage condition if its columns cover the whole set of possible patterns at least twice, i.e.,*

$$\forall x \in \{0,1\}^m \ \exists i_1 \neq i_2 \in [d] : \mathbf{V}_{:,i_1} = \mathbf{V}_{:,i_2} = x. \qquad \text{(coverage)}$$

*If both conditions are satisfied for $\mathbf{U}$ and $\mathbf{V}$, resp., then the adjacency matrix $\mathbf{A} = \mathbf{U} \diamond \mathbf{V}$ of the half-square graph $G$ has $2^m$ distinct columns (treated as binary vectors), and $\mathbf{A}$ has Boolean rank $m$. Moreover, for every entry $(i,j)$ of the adjacency matrix, the matrix $\mathbf{A}'$ obtained by replacing $(\mathbf{U} \diamond \mathbf{V})_{ij}$ by $1 - (\mathbf{U} \diamond \mathbf{V})_{ij}$ has Boolean rank $m + 1$.*

*Proof.* The reader will notice that (non-overlap) for $\mathbf{U}$ in this lemma is a matrix formulation of the hypothesis in lemma 4 for all sets $S_1, S_2 \subseteq [m]$, and that (coverage) for $\mathbf{V}$ is a direct reformulation of the hypothesis in lemma 5.

Let us note that one consequence of (non-overlap) is that two different patterns in the columns of $\mathbf{V}$ will incur different patterns in the columns of $\mathbf{A}$ (the Hamming distance is bounded away from zero). Also, as a consequence of (coverage), we know that each pattern occurs at least once, and therefore the matrix $\mathbf{A}$ has $2^m$ distinct columns. Because at least $m$ factors in a Boolean decomposition are necessary to express $2^m$ distinct column patterns, $\mathbf{A}$ is of Boolean rank $m$ [60].

To show the second claim, for a fixed entry of the adjacency matrix, let $(i,j)$ be its indices. We replace $(\mathbf{U} \diamond \mathbf{V})_{ij}$ by $1 - (\mathbf{U} \diamond \mathbf{V})_{ij}$ in $\mathbf{A}$ to create $\mathbf{A}'$. By (non-overlap), the $j$th column of $\mathbf{A}'$ is distinct from all other columns in $\mathbf{A}'$, since it differs in exactly one entry from all the other columns, but these columns have a Hamming distance of at least two from each other. Moreover, since every pattern occurs at least twice in $\mathbf{A}$ by (coverage), all $2^m$ original column patterns in $\mathbf{A}$ are still present in $\mathbf{A}'$. Therefore, $\mathbf{A}'$ has $2^m + 1$ distinct columns. By the same argument as above, at least $m + 1$ factors are necessary to express $\mathbf{A}'$ with a Boolean decomposition, and thus $\text{Rank}_{\mathcal{B}}(\mathbf{A}') = m + 1$. $\square$

## C.4 Proof of the main theorem

We introduce a random factor graph model over $\mathcal{G}_d^m$, inspired by the Erdős-Rényi model [36]:

**Definition 8.** *(Random sequence of growing $f$-DiGraphs) Let $(G_{f,d})_{d=0}^{\infty}$ denote a random sequence of factor directed graphs defined recursively, with $G_{f,0} = (\varnothing, F, \varnothing)$. Let $G_{f,d} = (V_d, F, E_d)$ be the graph at step $d$. We define $V_{d+1} = V_d \cup \{v_{d+1}\}$ where $v_{d+1}$ denotes a new variable node. To obtain $E_{d+1}$, we connect the new variable node into and out of each factor independently with probability $p \in (0,1)$. Finally, we define $G_{f,d+1} = (V_{d+1}, F, E_{d+1})$ and have $G_{d+1} = G_{f,d+1}^2[V] \in \mathcal{G}_{d+1}^m$.*

The reader will note that those graphs are not necessarily acyclic, but our result for the Boolean rank is true for the more general class of $f$-DiGraphs under this model. We now introduce our toy model for edge perturbations.

**Definition 9.** *(Stochastic edge operator) Let $G \in \mathcal{G}_d^m$. We denote by $\Lambda$ the stochastic operator that samples a random entry of the adjacency matrix $\mathcal{A}(G)$ and either removes the present edge, or adds the absent edge.*

**Theorem 1** (Boolean rank instability for edge perturbation). *Let $G \in \mathcal{G}_d^m$ be sampled according to an Erdős-Rényi random directed graph model. The probability that adding or removing an edge increases the Boolean rank is arbitrarily high for large $d$:*

$$\mathbb{P}(Rank_{\mathcal{B}}(\Lambda(G)) > Rank_{\mathcal{B}}(G)) \geq 1 - \alpha q^d, \qquad (5)$$

*where $Rank_{\mathcal{B}}$ denotes the Boolean rank, and $\alpha > 0$ and $q \in (0,1)$ depend on $m$ and the parameters of the random graph model.*

*Proof.* Let us denote the rank increase event by $R = \{\text{Rank}_{\mathcal{B}}\left(\Lambda(G_d)\right) > \text{Rank}_{\mathcal{B}}\left(G_d\right)\}$. Thanks to Lemma 6, if the random matrices $\mathbf{U}$ and $\mathbf{V}$ corresponding to $G$ fulfill (non-overlap) and (coverage), the rank increases no matter which edge is flipped due to $\Lambda$. Therefore, we have

$$\mathbb{P}(\bar{R}) \leq \mathbb{P}\left(\{\text{not (non-overlap)}\} \text{ or } \{\text{not (coverage)}\}\right) \tag{59}$$

$$\leq \mathbb{P}\left(\{\text{not (non-overlap)}\}\right) + \mathbb{P}\left(\{\text{not (coverage)}\}\right) \tag{60}$$

$$\leq \mathbb{P}\left(\bigcup_{S_1, S_2} \{|(\cup_{j \in S_1} \mathcal{U}_j) \triangle (\cup_{j' \in S_2} \mathcal{U}_{j'})| \leq 1\}\right) + \mathbb{P}\left(\{\text{not (coverage)}\}\right) \tag{61}$$

$$\leq 2^{2m} \mathbb{P}\left(\text{for fixed } S_1, S_2 : |(\cup_{j \in S_1} \mathcal{U}_j) \triangle (\cup_{j' \in S_2} \mathcal{U}_{j'})| \leq 1\right) + \mathbb{P}\left(\{\text{not (coverage)}\}\right) \tag{62}$$

$$\leq 2^{2m} \gamma q_A^d + \delta q_B^d \tag{63}$$

$$\leq \left(2^{2m} \gamma + \delta\right) \left(\max\{q_A, q_B\}\right)^d, \tag{64}$$

where we used union bounds and Lemmas 4 and 5 to bound the probabilities. Folding the linear dependence of $\gamma$ and $\delta$ on $d$ into the exponential term $q^d$ then concludes the proof. $\qquad\square$

### C.5 Empirical validation

We simulated $f$-DAGs using the same methodology presented in Appendix B.3 with $p_v = p_f = 0.5$.

**Boolean rank** For each combination of $d \in \{10, 20, \ldots, 90, 100\}$ and $m \in \{5, 10\}$, we simulated $T = 2$ $f$-DAGs. In the first simulation, we added an edge at random. In the second, we removed an edge at random. In Figure 2B, we calculated the Boolean rank of the adjacency matrix of the half-square graph $G$ before and after perturbation, using the methodology and the code from [40] to approach the NP-hard problem of Boolean matrix factorization. More precisely, [40] provide a solver for the best Boolean rank $\tilde{m}$ approximation of a matrix $\mathbf{A}$ in Frobenius norm. Starting from the linear rank of the matrix $\mathbf{UV}$ as an initial guess, we repeatedly used this solver to obtain the smallest $\tilde{m}$ that resulted in a Frobenius norm residual of 0, i.e., perfect matrix reconstruction using $\tilde{m}$ factors. After edge perturbation, we re-ran the solver and classified the graph as leading to a rank increase if the residual was 1 and as not leading to a rank increase if the residual was 0. Intuitively, an error of 1 means that the new edge cannot be described by any set of $\tilde{m}$ factors, and an error of 0 means that a (potentially different) factorization with $\tilde{m}$ factors reconstructs the perturbed graph.

Out of the 400 simulations, there were two configurations where the resulting residual was neither 0 nor 1. In both cases, the algorithm had not converged in time, and we excluded these two runs from the analysis. We found that the fraction of these suboptimal factorizations increased rapidly for larger $m$ and therefore limited the analysis to small values of $m$.

**Matrix rank** For larger values of $m$, we calculated the linear rank of the (binary) adjacency matrix of the perturbed graph. We note that the linear rank is only a heuristic approximation of the quantitative increase in complexity of the underlying matrix, because in general it provides neither an upper nor a lower bound on the Boolean rank [41]. In this scenario, we chose a ratio of edges to perturb in the half-square graph $G = (V, E)$, given by the target False Discovery Rate (FDR) $q$. Out of these $q|E|$ corruptions, we sampled a number $N_1 = \text{Binomial}(q|E|, 1/2)$ of edges present in $G$ uniformly at random to remove from the graph, and $N_2 = q|E| - N_1$ of edges not present in $G$ uniformly at random to add. Then, we reported the linear rank before and after perturbations for $T = 200$ configurations for every target FDR.

## D Characterization of acyclicity

We start by proving the graphical rule for acyclicity emerging from the structure of the bipartite graph.

**Lemma 1** (Induced acyclicity). *Let $G = (V, F, E)$ be an $f$-DiGraph. Then,*

$$G_f \text{ is acyclic} \iff G = G_f^2[V] \text{ is acyclic} \iff G_f^2[F] \text{ is acyclic.} \tag{6}$$

*Proof.* To prove this, by symmetry between the sets of nodes $F$ and $V$, it is enough to prove the first equivalence. We prove each implication of the first equivalence by contraposition.

Let us first assume that there exists a cycle in $G_f$, that is, that there exists a path starting and ending at the same node. That node is either a feature node or a factor node. If the node is a feature node $v$, then we may write the path as $[(v, f_{i_1}), (f_{i_1}, v_{j_1}), \ldots, (v_{j_{\omega-1}}, f_{i_\omega}), (f_{i_\omega}, v)]$. By definition of the half-square, all feature nodes in this path are directly connected by at least one factor node. Therefore, the sequence $[(v, v_{j_1}), \ldots (v_{j_{\omega-1}}, v)]$ is a path in $G$. However, this path links $v$ to itself, so it is a cycle. If the node is a factor, we may simply shift the cycle one step to obtain a path that starts from a feature node.

Let us now assume that there exists a cycle in $G$, for example $[(v, v_{j_1}), \ldots (v_{j_{\omega-1}}, v)]$. For every edge in $G$, there is by definition an edge of length 2 in $G_f$. Consequently, for every edge $(v_{j_a}, v_{j_{a+1}})$ of the path in $G$, there exists a path of length 2 $(v_{j_a}, f), (f, v_{j_{a+1}})$ in $G_f$ connecting $v_{j_a}$ and $v_{j_{a+1}}$. By concatenating these paths, we obtain a cycle in $G_f$. □

We also note the existence of a more algebraic proof, based on the two following arguments. First, notice that for a matrix $W \in \mathbb{R}_+^{n \times n} = UV$, where all entries of $U$ and $V$ are also non-negative, $W$ is nilpotent if and only $W$ is a DAG. Second, $VU$ is nilpotent if and only if $UV$ is nilpotent.

## D.1 Trace-exponential characterization

Interestingly, in the case of an $f$-DiGraph $G_f$, we can be more precise, and prove that the tr-exp penalty applied to $G$ can be related to the one applied to $G_f^2[F]$.

**Proposition 6** ($\operatorname{Tr}\exp$ penalty on $f$-DiGraphs). *Let $G_f = (V, F, E)$ be an $f$-DiGraph, and $(\mathbf{U}, \mathbf{V})$ its partial adjacency matrices. Then, the two following quantities are identical,*

$$\operatorname{Tr}\exp\{\mathbf{UV}\} - d = \operatorname{Tr}\exp\{\mathbf{VU}\} - m, \tag{65}$$

*and are both equal to zero if and only if $G_f$ or, equivalently, any of its half-squares, is acyclic.*

*Proof.* Because $\mathbf{UV}$ is a positively weighted adjacency matrix for $G$, we have that $\operatorname{Tr}\exp\{\mathbf{UV}\} = d$ if and only if $G$ is acyclic [19]. Similarly, $\operatorname{Tr}\exp\{\mathbf{VU}\} = m$ if and only if $G_f^2[F]$ is acyclic. Considering the equivalences of acyclicity in Proposition 1, we must only prove the algebraic identity. For this, we write

$$\operatorname{Tr}\exp\{\mathbf{UV}\} - d = \operatorname{Tr}\left[\sum_{k=0}^{\infty} \frac{(\mathbf{UV})^k}{k!}\right] - d \tag{66}$$

$$= \sum_{k=1}^{\infty} \frac{\operatorname{Tr}\left[(\mathbf{UV})^k\right]}{k!} \tag{67}$$

$$= \sum_{k=1}^{\infty} \frac{\operatorname{Tr}\left[\mathbf{U}(\mathbf{VU})^{k-1}\mathbf{V}\right]}{k!} \tag{68}$$

$$= \sum_{k=1}^{\infty} \frac{\operatorname{Tr}\left[(\mathbf{VU})^{k-1}\mathbf{VU}\right]}{k!} \tag{69}$$

$$= \sum_{k=1}^{\infty} \frac{\operatorname{Tr}\left[(\mathbf{VU})^k\right]}{k!} \tag{70}$$

$$= \operatorname{Tr}\exp\{\mathbf{VU}\} - m, \tag{71}$$

where we made use of the fact that $\operatorname{Tr}(\mathbf{AB}) = \operatorname{Tr}(\mathbf{BA})$. □

## D.2 Spectral characterization

We present here a version of the power iteration method tailored to the case of $f$-DiGraphs. It calculates an approximation to the spectral radius of $\mathbf{UV}$, in turn approximately characterizing acyclicity of the corresponding graph $G$ (as presented in [26]).

---

**Algorithm 1** Factor power iteration

---

**input** Factor adjacency matrices $(\mathbf{U}, \mathbf{V}) \in \mathbb{R}^{d \times m} \times \mathbb{R}^{m \times d}$ and number of iterations $T \in \mathbb{N}$
  1: Initialize $p_0 \in \mathbb{R}^d$ and $q_0 \in \mathbb{R}^d$, either randomly or by warm-starting with previous estimates of the leading singular vectors of $\mathbf{UV}$
  2: **for** $t \in \{0, \ldots, T-1\}$ **do**
  3:   $p_{t+1} = \mathbf{V}^\top \mathbf{U}^\top p_t / \|\mathbf{V}^\top \mathbf{U}^\top p_t\|_2$
  4:   $q_{t+1} = \mathbf{UV} q_t / \|\mathbf{UV} q_t\|_2$
**output** Approximate leading singular value $\hat{\rho} = \frac{p_T^\top \mathbf{UV} q_T}{p_T^\top q_T}$

---

# E   Implementation details for DCD-FG

In this section, we present the implementation details for DCD-FG.

## E.1   Factor MLP: Likelihood model and architecture details

For adjacency matrices $\mathbf{U}$ and $\mathbf{V}$ of the $f$-DiGraph $G_f$, we detail now how, for DCD-FG, we construct a likelihood model that is both non-linear and uses the factor semantics. We define $h_f$ as a (deterministic) variable attached to factor node $f \in F$, calculated as the scalar output of a multi-layer perceptron (MLP) on the input variables of each factor defined by the matrix $\mathbf{U}$,

$$h_f = \text{MLP}(\mathbf{U}_{:,f} \circ X; \Theta_f), \tag{72}$$

for neural networks parameters $\Theta_f$, and $X \in \mathbb{R}^d$, where masking with $\mathbf{U}$ is a computationally efficient way to restrict the input to the neural network to a potentially varying set of input variables. We provide the default hyperparameters for these MLPs in Appendix F. Further, we let the conditional distribution of each node depend linearly on its parent factors defined by the matrix $\mathbf{V}$,

$$X_j \sim \text{Normal}\left(\alpha_j^\top(\mathbf{V}_{j,:} \circ h) + \beta_j, \sigma_j^2\right), \tag{73}$$

for parameters $\alpha_j \in \mathbb{R}^m$, $\beta_j \in \mathbb{R}$, $\sigma_j > 0$.

These equations correctly specify a density model $p_\Theta(X_j \mid X_{-j})$ as long as the obtained likelihood does not depend on feature $X_j$, but only on the other features $X_{-j}$. In particular, this is true for all $j$ if $\text{diag}(\mathbf{UV}) = \mathbf{0}_{d \times d}$ (i.e., if there are no self-loops in $G$).

## E.2   Linear model and relationship to NOTEARS-LR

We briefly explore connections between our factor architecture, and the NOTEARS-LR model [20]. In the context of our factor architecture with linear models, each factor variable $h_f$ is a linear combination of $X$ with weights $r_f \in \mathbb{R}^d$. Omitting bias terms for simplicity, we collect these weights into a matrix $\mathbf{R} = [r_1, \ldots, r_m]$. Similarly, each feature $X_j$ is a linear combination of $h_f$ with weights $\alpha_j \in \mathbb{R}^m$ that we similarly condense into a matrix $\mathbf{A} = [\alpha_1, \ldots \alpha_d]$. In this case, the mean of our Gaussian conditional likelihood model can be written as

$$\mathbb{E}[X_j \mid X] = \mathbf{AR}X. \tag{74}$$

Interestingly, without further modifications, we observed that this model suffers from mis-specification, as well as poor performance due to the possibility of self-loops. In the linear case, a simple workaround to this problem is to mask contributions due to self-loops by removing the diagonal of the matrix $\mathbf{AR}$. This corresponds to modifying the likelihood model to

$$\mathbb{E}[X_j \mid X] = \mathbf{M}_{\text{no-loop}} \circ (\mathbf{AR})X, \tag{75}$$

where $\mathbf{M}_{\text{no-loop}} = \mathbb{1}\mathbb{1}^\top - \mathbf{I}_{d \times d}$ is a self-loop removing mask. This modification is exactly the linear model proposed by NOTEARS-LR. While this simple modification effectively eliminates self-loops, it necessitates expanding the full matrix $\mathbf{AR}$, which incurs a runtime cost of $O(d^2)$. Next, we describe an alternative approach that avoids this step.

### E.3 Removing self-loops in $f$-DAGs

Parametrizing the $f$-DAG adjacency matrix by $\mathbf{M} = [\mathbf{U}, \mathbf{V}]$ with otherwise unconstrained binary matrices $\mathbf{U}$ and $\mathbf{V}$ causes the induced feature graphs to potentially have a large number of self-loops. As emphasized before, this causes the local likelihood to not be correctly specified. We also found this to be detrimental to the performance of the model, even though the acyclicity score should promote the absence of self-loops in later stages of training. We hypothesize that when self-loops are present, the model starts off training by predicting every variable by itself, which in turn prevents any other signal from being picked up during the training stage. Therefore, the model never learns a meaningful predictive model that can be accurately pruned into a DAG.

To circumvent this issue, we propose an alternative model in which the matrices $\mathbf{U}$ and $\mathbf{V}$ are additionally constrained to explicitly remove self-loops. More precisely, we calculate $\mathbf{M} = [\mathbf{U}, \mathbf{V}]$ as a deterministic mapping from a single $d \times m$ matrix $\mathbf{W}$, taking value in $\{0, -1, 1\}^{d \times m}$ with independent entries sampled according to a Gumbel-softmax distribution [61] with parameters $\Phi$.

The intuition behind this single matrix is that each individual entry $\mathbf{W}_{ij}$ decides whether there is an edge between a factor $f_j$ and a feature $v_i$, as well as its orientation. More precisely,

$$\forall i \in [d], j \in [m], (\mathbf{U}_{ij}, \mathbf{V}_{ji}) = \begin{cases} (1, 0), & \text{for } \mathbf{W}_{ij} = 1, \\ (0, 1), & \text{for } \mathbf{W}_{ij} = -1, \\ (0, 0), & \text{for } \mathbf{W}_{ij} = 0. \end{cases} \tag{76}$$

Because the entries $\mathbf{U}_{ij}$ and $\mathbf{V}_{ji}$ may never be both equal to 1, there are no self-loops in the induced half-square graph. Indeed, the number of self-loops is simply the trace of the adjacency matrix, which is equal to zero,

$$\mathrm{Tr}(\mathbf{UV}) = \sum_{i=1}^{d} \sum_{j=1}^{m} \mathbf{U}_{ij} \mathbf{V}_{ji} = 0. \tag{77}$$

In the case of DCD-FG, it is important to notice that the matrices $\mathbf{U}$ and $\mathbf{V}$ are not fixed, but sampled from a random distribution $\mathbf{M}(\Phi)$ (we drop dependence to the parameter $\Phi$ for convenience of notation). For the purpose of later enforcing acyclicity, we calculate the expectation of the weighted (random) adjacency matrices formed from $\mathbf{M}(\Phi)$, which, by independence of the entries of $\mathbf{W}$, is

$$\forall i, j \in [d]^2, \mathbb{E}[\mathbf{UV}]_{ij} = \begin{cases} (\mathbb{E}[\mathbf{U}]\mathbb{E}[\mathbf{V}])_{ij}, & \text{for } i \neq j, \\ 0, & \text{for } i = j. \end{cases} \tag{78}$$

Analogously, the expectation of the adjacency matrix of the induced factor graph is

$$\forall k, \ell \in [m]^2, \mathbb{E}[\mathbf{VU}]_{k\ell} = \begin{cases} (\mathbb{E}[\mathbf{V}]\mathbb{E}[\mathbf{U}])_{k\ell}, & \text{for } k \neq \ell, \\ 0, & \text{for } k = \ell. \end{cases} \tag{79}$$

### E.4 Acyclicity penalties

We now show how acyclicity penalties can be applied to our factored representations (tr-exp factor and spectral factor in the experiments), and that they have the claimed runtime bounds, i.e., that they are linear in $d$. For both the tr-exp and the spectral radius penalty, we apply the continuous acyclicity penalty $\mathcal{C}(\mathbb{E}[\mathbf{M}(\Phi)])$ to the expectation of $\mathbf{M}(\Phi) = [\mathbf{U}(\Phi), \mathbf{V}(\Phi)]$, as in [6]. We leave the investigation of instead calculating gradients of the expected penalty $\mathbb{E}[\mathcal{C}(\mathbf{M}(\Phi))]$ with respect to $\Phi$ using reparameterized samples of $\mathbf{M}(\Phi)$ as future work.

**Tr-exp factor penalty**    We apply the tr-exp penalty to $\mathbb{E}[\mathbf{VU}]$, the weighted adjacency matrix on the half-square $G_f^2[F]$ induced by the factor nodes. As described above in (79), $\mathbb{E}[\mathbf{VU}]$ is calculated by calculating the matrix product of the expectations of $\mathbf{V}$ and $\mathbf{U}$, before setting the diagonal to zero. This $m \times m$ matrix can be calculated in $O(m^2 d)$, and the gradient of the penalty $h(\mathbb{E}[\mathbf{M}]) = \mathrm{Tr}\exp\{\mathbb{E}[\mathbf{VU}]\} - m$ can be calculated in $O(m^3)$, yielding a total runtime of $O(m^3 + m^2 d)$.

**Spectral radius factor penalty**    We apply the factor power iteration (Algorithm 1) to the matrix $\mathbb{E}[\mathbf{UV}]$ to maintain left and right eigenvectors for the leading singular value in $O(Tmd)$ operations, and then calculate the gradient of that singular value in time $O(md)$. In this case, we use a simple modification of Algorithm 1 with a diagonal offset, noticing that

$$\mathbb{E}[\mathbf{UV}] = \mathbb{E}[\mathbf{U}]\mathbb{E}[\mathbf{V}] - \mathrm{diag}(\mathbb{E}[\mathbf{U}]\mathbb{E}[\mathbf{V}]), \tag{80}$$

and that the matrix-vector multiplications in Algorithm 1 can be calculated in time $O(md)$.

### E.5 Augmented Lagrangian

This section is adapted from the supplementary materials of the work of Brouillard, Lachapelle et al. [6] and outlines how the DAG-constrained optimization problem can be solved with first-order methods.

Let us recall that the score function and the optimization problem for DCD-FG, assuming perfect interventions, are defined as:

$$\max_{\Phi,\Theta} \mathcal{S}(\Phi,\Theta) \text{ such that } \mathcal{C}(\mathbb{E}[\mathbf{M}(\Phi)]) = 0, \tag{81}$$

$$\text{where } \mathcal{S}(\Phi,\Theta) = \mathbb{E}_{\mathbf{M}'\sim\mathbf{M}(\Phi)}\left[\sum_{k=1}^{K}\mathbb{E}_{X\sim P_{\text{data}}^{(k)}}\sum_{j\notin\mathcal{I}_k}\log p_{\Theta}^{j}(X_j;\mathbf{M}_j',X_{-j})\right] - \lambda\left\|\mathbb{E}[\mathbf{M}(\Phi)]\right\|_1. \tag{82}$$

The augmented Lagrangian transforms the constrained problem into a sequence of unconstrained problems of the form

$$\max_{\Phi,\Theta} \mathcal{S}(\Phi,\Theta) - \gamma_t\mathcal{C}(\mathbb{E}[\mathbf{M}(\Phi)]) - \frac{\mu_t}{2}\left(\mathcal{C}(\mathbb{E}[\mathbf{M}(\Phi)])\right)^2, \tag{83}$$

where $\gamma_t$ and $\mu_t$ are the Lagrange multiplier and the penalty coefficient of the $t$-th unconstrained optimization problem, respectively. In all our experiments, we initialize $\gamma_0 = 0$ and $\mu_0 = 10^{-8}$. Each such problem is approximately solved using a first-order stochastic optimization procedure (RMSProp in our experiments). We assume that a subproblem has converged when (83) evaluated on a validation set stops increasing. Let $(\Phi_t^*,\Theta_t^*)$ be the approximate solution to subproblem $t$. Then, $\gamma_t$ and $\mu_t$ are updated according to the following rule:

$$\gamma_{t+1} = \gamma_t + \mu_t\mathcal{C}(\mathbb{E}[\mathbf{M}(\Phi_t^*)]) \tag{84}$$

$$\mu_{t+1} = \begin{cases} \eta\mu_t, & \text{if } \mathcal{C}(\mathbb{E}[\mathbf{M}(\Phi_t^*)]) > \delta\mathcal{C}(\mathbb{E}[\mathbf{M}(\Phi_{t-1}^*)]), \\ \mu_t, & \text{otherwise,} \end{cases} \tag{85}$$

with $\eta = 2$ and $\delta = 0.9$. Each subproblem $t$ is initialized using the previous subproblem's solution $(\Phi_t^*,\Theta_t^*)$. The augmented Lagrangian procedure is stopped when $\mathcal{C}(\mathbb{E}[\mathbf{M}(\Phi_t^*)]) < 10^{-8}$, or $\mu_t > 10^{32}$.

The gradient of (83) with respect to the parameters $\Phi$ and $\Theta$ is estimated on a minibatch of observations. To compute the gradient of the likelihood part with respect to $\Phi$, we follow [6] and use a Straight-Through Gumbel-Softmax estimator [61]. This approach relies on discrete Bernoulli samples at the forward pass, but uses the reparameterized samples of the Gumbel softmax distribution for the backward pass (with fixed temperature parameter $T = 1$).

### E.5.1 Code Statement

We implemented DCD-FG in PyTorch, using the DCDI codebase as a starting point. We extensively refactored and modified the original code (simulations, loss functions, training, evaluation) in order to scale to thousands of variables, and to accommodate the simulation of factor graphs. The software is available as open-source on GitHub at `https://github.com/Genentech/dcdfg` under the Apache 2.0 licence.

## F Details for empirical evaluation of DCD-FG

In this section, we provide the necessary details for reproducing the experiments in the paper.

### F.1 Synthetic data sets

For each type of synthetic data set, we first sampled an $f$-DAG as explained in Appendix B.3, with $p_v = 0.1$ and $p_f = 0.2$ and then we sampled the causal mechanisms, adapting the method from [6] as follows. In each of our half-square graphs ($d = 100$), and for each intervention regime $k \in [K]$, where $K = 100$, intervention targets with a size of 1 to 3 nodes were chosen uniformly at random. $n/(K+1)$ independent observations were sampled for each interventional setting. The data were normalized, i.e., we subtracted the mean and divided by the standard deviation. In cases where IGSP

required feature aggregation, we clustered the features using spectral clustering as implemented by scikit-learn [62].

In the linear data sets, each node was set to be a linear function of its parent nodes (in the $f$-DAG), with additional Gaussian noise of standard deviation $\sigma = 0.4$. The coefficients were sampled uniformly from $[-1, -0.25] \cup [0.25, 1]$ (to make sure they are bounded away from zero). Interventions were handled by instead sampling the intervened-upon node from an isotropic Gaussian distribution with unit variance.

In the non-linear data sets (NN), each node was set to be a non-linear function of its parents nodes (in the $f$-DAG), with additional Gaussian noise of standard deviation $\sigma = 0.4$. The non-linear functions were fully-connected neural networks with one hidden layer of 20 units and hyperbolic tangent as nonlinearitiy in the hidden layer. The weights of each neural network were sampled from isotropic Gaussian distributions with unit variance. Similarly to the linear model, interventions were handled by instead sampling the intervened-upon node from an isotropic Gaussian distribution with unit variance.

### F.2 Preprocessing of the Perturb-CITE-seq data set

We downloaded the data set from the Single Cell Portal of the Broad Institute (accession code SCP1064). We converted the data from log-normalized count per millions into raw counts for processing with the scanpy package [63]. We removed cell profiles with less than 500 expressed genes, and genes expressed in less than 500 cells. Then, we filtered the genes to include only the genetically-perturbed genes and the most variable genes for a total of $d = 1,000$ genes. We finally partitioned the cells from each of the three conditions (co-culture, IFN$\gamma$, and control) into distinct datasets. We used spectral clustering as implemented in sklearn for clustering, and selected three gene sets with $10, 20$ and $50$ modules.

### F.3 Baseline Methods

In this section, we provide additional details on the baseline methods and cite the implementations that were used. NOTEARS [19] was extended to handle perfect interventions, and to use a Gaussian likelihood (with unequal variance across features). In contrast to the original implementation that used a second-order optimization method, the reimplementation used in this paper relies on first-order optimization. A similar GPU reimplementation was used in the NO-BEARS manuscript [26] and shown to be 100x faster for $d = 300$. NOTEARS-LR [20] and NOBEARS [26] were also reimplemented to handle interventions, and use a Gaussian likelihood.

We noticed floating point overflow in most experiments ($d > 100$) using 16 bit precision when calculating the tr-exp penalty. We first explored using 32 bit precision, but instead preferred scaling the matrix before computation of the penalty. We have noticed that using the spectral radius of the initialized adjacency matrix as scaling factor provided a nice safeguard, and have applied this throughout all the models.

Finally, we noticed that for the NOTEARS baseline, thresholding the weight matrix at $w^* = 0.3$ (as done in the original NOTEARS paper) provided poor performance on this benchmark, and in the biological dataset. Therefore, we adopted a single strategy for pruning the adjacency matrices into DAGs for all of NOTEARS, NOBEARS, NOTEARS-LR and DCD-FG. We threshold the weighted adjacency matrices (weights for NOTEARS, NOBEARS and NOTEARS-LR, and probabilities of edge for DCD-FG) where the threshold $t^*$ is obtained by binary search with $T = 20$ evaluation of an (exact) acyclicity test to find the largest possible DAG for each method.

For IGSP, we used the implementation from `https://github.com/uhlerlab/causaldag`. The cutoff values used for `alpha-inv` was always the same as `alpha`. We used tools from the Python package Causal Discovery Toolbox [45] for calculating the SHD metrics.

### F.4 Default hyperparameters and hyperparameter search

For all methods, we performed an exhaustive hyperparameter grid search. The models were trained on $80\%$ observations and evaluated on $20\%$ of the remaining ones (distinct interventions were used in the training and the held-out data used for evaluations). For all of NOTEARS, NOBEARS, NOTEARS-LR and DCD-FG, we searched over the regularization coefficient for sparsity $\lambda$. Additionally, for

NOTEARS-LR and DCD-FG, we searched over the number of learned factors $m$. For DCD-FG, we used the acyclicity penalty as a supplementary hyperparameter (spectral or trace of exponential). For IGSP, we scored the output by using it as a mask for fitting a linear model, and we explored several cutoff values `alpha` based on the value used in the original publication [7], and the best performing value in the experiments of [6]. Because IGSP did not terminate in many scenarios (after 5 hours, even with the Gaussian conditional independence), we ran the method in several instances of feature aggregation and observation subsampling and report the best performance. Feature aggregation means that we decreased the dimensionality of the dataset by summarizing the feature set into a cluster set by averaging all features within a cluster, and mapping intervention targets from the feature set to the cluster set. The complete hyperparameter search space for each algorithm is described in Table 2.

Table 2: Hyperparameter search spaces for each algorithm.

| | Hyperparameter space |
|---|---|
| **DCD-FG** | $\log_{10}(\lambda) \in \{-3, -2, -1, 0, 1, 2\}$ 
 $m \in \{10, 15, 20, 30, 50\}$ 
 DAG penalty $\in \{\text{spectral, tr-exp}\}$ |
| **NOTEARS-LR** | $\log_{10}(\lambda) \in \{-3, -2, -1, 0, 1, 2\}$ 
 $m \in \{10, 15, 20, 30, 50\}$ 
 DAG penalty $\in \{\text{spectral, tr-exp}\}$ |
| **NOTEARS** | $\log_{10}(\lambda) \in \{-3, -2, -1, 0, 1, 2\}$ 
 DAG penalty $\in \{\text{spectral, tr-exp}\}$ |
| **NOBEARS** | $\log_{10}(\lambda) \in \{-3, -2, -1, 0, 1, 2\}$ |
| **IGSP** | `alpha` $\in \{1e-3, 1e-5\}$ 
 CI test $\in \{\text{Gaussian, KCI}\}$ 
 Feature clustering (# clusters) $\in \{10, 20, 50\}$ 
 Observation sub-sampling (only for bio dataset) $\in \{0.1, 0.25, 0.5\}$ |

DCD-FG, NOTEARS-LR, NOTEARS and NOBEARS share several default hyperparameters related to the optimization procedure and the architecture of the neural networks (for DCD-FG only) that we outline in Table 3 (values are similar to those in [6]). Neural networks were designed with leaky-ReLU activation functions, and initialized following the Xavier initialization [64]. RMSprop was used as the optimizer [65] with minibatches of size 64.

Table 3: Default hyperparameters for DCD-FG, NOTEARS, NOTEARS-LR and NOBEARS.

| Hyperparameters |
|---|
| $\mu_0 = 10^{-8}$, $\gamma_0 = 0$, $\eta = 2$, $\delta = 0.9$ |
| Augmented Lagrangian constraint threshold: $10^{-8}$ |
| Learning rate: $2.10^{-3}$ |
| # hidden units: 16 (DCD-FG only) |
| # hidden layers: 2 (DCD-FG only) |

### F.5 Assessment of statistical significance

For every claim of the form "Algorithm X and Y outperformed Algorithm W and Z with respect to metric $h$", we applied a Wilcoxon signed-rank test to the difference of scores between the worst performing model out of X, Y and the best performing model out of W, Z. We alternatively applied an Mann–Whitney U test (unpaired test) by concatenating the results of each algorithm (X, Y) and (W, Z) and recovered identical statistical significance assessments.

### F.6 Additional experimental results

Here, we report the results of additional experiments intended to complement the experiments presented in the main paper and to investigate the robustness of DCD-FG:

- Aggregation of precision and recall on our Gaussian causal structural models, as presented in Figure 4, into a F1-score (Figure 7)

- Performance of DCD-FG for different values of the rank parameter ($m$) (Figure 8) on Gaussian causal structural models. For this, we considered the same data as for Figure 4, but performed hyperparameter search over all parameters besides $m$, for different choices of $m$. We observe that the performance is fairly robust to a wide variety of choices of $m$, and that validation likelihood is strongly correlated with the resulting performance, justifying our selection of $m$ by optimizing hold-out likelihood.

- Alternative benchmark for a different exogenous noise distribution (uniform instead of Gaussian; with matching variance; Figure 9). For this, we chose the same graph generative and conditional likelihood model as in Figure 4, only changing the noise distribution. While performance slightly degrades, the relative ordering of DCD-FG compared to NOTEARS-LR and NOTEARS remained the same.

- Alternative benchmark with observational data (Figure 10). Using the same graph generative and conditional likelihood model as in Figure 4, we generated the same number of observations, but without any interventions. Again, while performance decreases in this regime, DCD-FG remains the best-performing method.

- Total runtime of methods on all experiments (Figure 11).

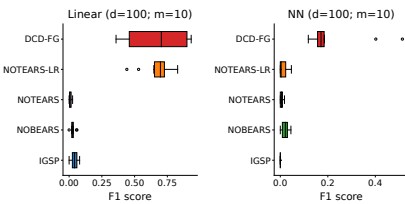

Figure 7: F1-score for Gaussian causal structural models experiments.

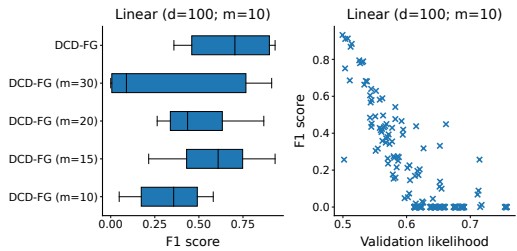

Figure 8: Robustness of DCD-FG to the rank hyperparameter.

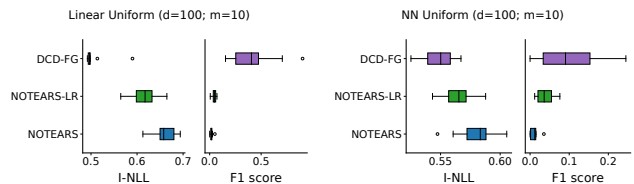

Figure 9: Robustness of DCD-FG to the model mispecification.

### F.7 Biological interpretation of the $f$-DAG

The $f$-DAG inferred by DCD-FG for the cells treated with IFN$\gamma$ has $m = 20$ factors, and the half-square $G$ has 196,303 edges. We show a histogram of ingoing and outgoing edges to each factor in Figure 12.

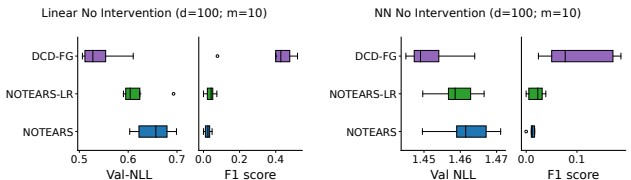

Figure 10: Performance of DCD-FG with observational data.

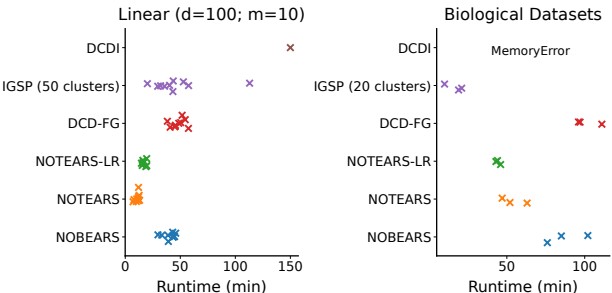

Figure 11: Total runtime of DCD-FG on one NVIDIA Tesla T4 GPU with 15Gb of RAM.

In order to assign biologically meaningful names to the factors, we performed Gene Set Enrichment Analysis (GSEA) via the enrichr method [50], applied independently to each factor $f$ by considering the set of genes that are connected to $f$ in either direction. In order to place the genes onto the factor half-square, we inferred for each gene the strongest factor-to-factor edge in which it appears. Indeed, we noticed that the same gene may appear upstream and/or downstream of several factors. For better interpretability, we selected the strongest parent factor and child factor based on the weights of the model (either the linear model in case of link from factor to gene, or the first layer of the MLP in case of link from gene to factor). We anticipate that further work will be necessary to visualize and interpret those large graphs for more involved biological applications, but we currently leave this as future work.

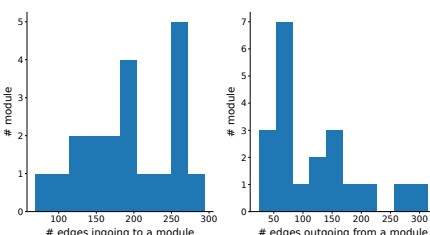

Figure 12: Distributions of ingoing and outgoing edges of modules in the $f$-DAG estimated on the IFN$\gamma$ treated cells.