# OpenReview forum: "Large-Scale Differentiable Causal Discovery of Factor Graphs"
_NeurIPS.cc/2022/Conference — NeurIPS 2022 Accept_

### Official Review · Reviewer_WAgt · 2022-07-08

**Rating:** 6
**Confidence:** 3
**Soundness:** 4 excellent
**Presentation:** 3 good
**Contribution:** 3 good

**Summary:**

This paper aims to solve the large-scale causal discovery problem with hundreds of nodes. To this end, the authors first introduce factor directed acyclic graphs (f-DAGs) which have many promising properties by incorporating the structural assumption to restrict the search space and reduce the optimization difficulty. Then, they propose Differentiable Causal Discovery of Factor Graphs (DCD-FG) to complete large-scale causal discovery. Theoretical analyses and experimental results demonstrate the superiority of f-DAGs and DCD-FG.

**Questions:**

1. How to justify that the low rank assuption is commonly met in real scenarios?

2. Although intuitively making sense, is there a theoretical result for the claim that "more statistically robust in the high-dimensional regime where the underlying skeleton is hard to assess"?

3. It seems most experimental results suround the SHD, and F1 score. Since your claim is fast searching, can you also show some computational efficiency results?

**Limitations:**

There are no  limitations and potential negative societal impact of their work.

**Strengths And Weaknesses:**

Strength:
1. This paper effectively reduce the search space in the large-scale causal discovery problem by introducing factor directed acyclic graphs. Based on which they propose Differentiable Causal Discovery of Factor Graphs to realize scalable causal discovery.

2. The experiments are well designed. This paper provides runtime experiments, simulation studies, and a case study on single-cell RNA sequencing data with hundreds of genetic interventions to demonstrate the efficacy of the proposed method.

Weakness:
1. It seems quite confusing in line 41-44. In the first sentence, it says “sparsity [17] as well as low-rank assumptions [18] are often exploited in algorithms”. However, in the next sentence, it points out “the use of low-rank constraints remains largely under-explored for this purpose”.

2. The theoretic property of this work is not well presented. The proposition 1 is only a very special case of lpw rank models. It would be better to justify the method in a wider scenarios.

---

> ### Author Response · Authors · 2022-08-02
> **Reponse to reviewer (1/2)**
>
> We thank the reviewer for their review and are glad they approve of our experimental validation. Note that due to the character limit in OpenReview comments, our response is broken into multiple parts.
>
> **Confusing introduction:** We acknowledge that lines 41-44 were confusing and will revise them in the final version.
>
> **Presentation of theoretic properties:** We apologize for any confusion caused by the presentation of Proposition 1 and clarify below the relationship of the Boolean low-rank models we consider with general low-rank models.
>
> We argue that one could consider a variety of low rank graph models. In particular, the following three seem natural to us:
>
> \begin{align*}
> \mathcal{G}\_{\mathrm{lin}}^m &= \\{ G : G \text{ admits weighted adjacency matrix}~ W \in \mathbb{R}^{d \times d} \\\\
> & \quad \quad \text{ with } W=UV \text{ for } U \in \mathbb{R}^{d \times m}, V \in \mathbb{R}^{m \times d}\\}, \text{ the set of linear rank $\leq m$ graphs}
> \end{align*}
>
> \begin{align*}
> \mathcal{G}\_{\mathrm{lin,nonneg}}^m &= \\{ G : G \text{ admits weighted adjacency matrix } W \in \mathbb{R}\_{\geq 0}^{d \times d} \\\\
> & \quad \quad \text{ with } W=UV \text{ for } U \in \mathbb{R}\_{\geq 0}^{d \times m}, V \in \mathbb{R}\_{\geq 0}^{m \times d}\\}, \text{ the set of linear non-negative rank $\leq m$ graphs}
> \end{align*}
>
> \begin{align*}
> \mathcal{G}\_{\mathrm{bool}}^m &= \\{ G : G \text{ admits adjacency matrix } A \in \\{0,1\\}^{d \times d} \\\\
> & \quad \quad \text{ with } A=U \diamond V \text{ for } U \in \\{0,1\\}^{d \times m}, V \in \\{0,1\\}^{m \times d}\\}, \text{ the set of Boolean rank $\leq m$ graphs}
> \end{align*}
>
> Proposition 1 shows that the factor graphs we consider give rise to $\mathcal{G}^m_{\mathrm{bool}}$. One can show that $\mathcal{G}^m_{\mathrm{bool}} = \mathcal{G}^m_{\mathrm{lin,nonneg}} \subsetneq \mathcal{G}^m_{\mathrm{lin}}$. We think the reviewer is focusing on this last strict inclusion.
>
> We respectfully disagree with the reviewer's assessment that the set of Boolean low rank graphs $\mathcal{G}^m_{\mathrm{bool}}$ constitutes a "very special case" of low-rank models. The only other class that comes to mind is $\mathcal{G}^m_{\mathrm{lin}}$, which might seem more natural at first glance when considering linear structural equation models. However, the only graphs in $\mathcal{G}^m_{\mathrm{lin}} \setminus \mathcal{G}^m_{\mathrm{bool}}$ that are not captured by DCD-FG are those where the contributions of several factors cancel out precisely to produce zeros not expected by the sparsity pattern of $U$ and $V$, leading to a violation of faithfulness in the associated factor graph. We argue that those graphs are fairly unintuitive and we do not see a straightforward way of extending the factor semantics of our nonlinear models to these models.
>
> We will clarify this point in the Supplementary Materials of the final version of the paper.
>
> **Real life examples for low-rank assumption:** Our model is in particular motivated by biological applications. One hallmark of transcriptional regulation of gene expression is the presence of "regulons", where one or a set of proteins, called transcription factors (TFs), all co-regulate one set of target genes. Moreover, those TFs can themselves be regulated at the transcriptional level as part of earlier regulons or by shared upstream proteins that activate them, and those upstream proteins are often co-regulated at the transcriptional level as regulons themselves. Concordantly, a low-rank structure has been observed in the total transcriptomic effects of genetic perturbations, for example, see Figure 4A in [1]. Such patterns have been observed across a multitude of systems [1, 2, 3, 4, 5, 6].
>
> Moreover, we consider the low-rank assumption a reasonable approximation to restrict the model class, even if it might be violated in practice. The favorable results on the Perturb-Seq data in Section 5.2 corroborate this.
>
> To clarify this point, we will add more information on the appropriateness of the low-rank assumption to the discussion section of the paper.

---

> > ### Author Response · Authors · 2022-08-02
> > **Reponse to reviewer (2/2)**
> >
> > **Theoretical results supporting robustness to noise:** We agree with the reviewer that a fully-fledged analysis of our estimator would be interesting, but we consider this out of scope of the paper in its present form. We partially address this point in Theorem 1 by considering random corruptions of the underlying skeleton of the causal graph as a proxy for statistical noise and showing that such corruptions are likely to increase the Boolean rank of the graph. Moreover, we consider the favorable generalization performance of DCD-FG in Section 5 as empirical evidence toward the claim of increased noise tolerance.
> >
> > To address this question, we will correct the relevant sentence in the abstract to be more precise, as the current version is an overstatement.
> >
> > **Runtime comparisons:** A preliminary comparison of runtime for specific steps of the algorithm can be found in Figure 3 of the paper. To further clarify the superior runtime of DCD-FG, we will add information on total runtime to the paper on both of the simulated data, and the biological case study (https://anonymous.4open.science/r/rebuttal-neurips-D93C/runtime_total.pdf). We now briefly describe the results.
> >
> > On our synthetic datasets, with a relatively small number of nodes (d=100), DCD-FG has reasonable runtime (50 min) in comparison to other algorithms. In order to provide a fair comparison with another neural networks based method, we ran DCDI on one of the datasets and noticed that DCD-FG is indeed significantly faster (50 vs 150 min). Additionally, we wish to note that on this small dataset, the runtime of vanilla NOTEARS can appear to be competitive with low-rank approaches (NOTEARS-LR & DCD-FG). However, this is mostly caused by NOTEARS not learning any meaningful graph. In those cases, the augmented Lagrangian method will terminate faster since the algorithm finds a very sparse adjacency matrix, resulting in the very poor statistical performance observed in Section 5.1.
> >
> > On our large-scale biological dataset, with thousands of features, we were hard-pressed to find _any_ algorithms that (1) admit non-linear functional relationships between nodes and (2) terminate in under 24 hours (IGSP requires clustering of features in 20 groups at most to terminate within a day; DCDI yields memory errors). By contrast, DCD-FG terminates in about 2 hours in most cases.
> >
> > [1] Dixit, Atray, et al. "Perturb-Seq: dissecting molecular circuits with scalable single-cell RNA profiling of pooled genetic screens." Cell 167.7 (2016): 1853-1866.
> >
> > [2] Frangieh et al. "Multimodal pooled Perturb-CITE-seq screens in patient models define mechanisms of cancer immune evasion." Nature Genetics (2021)
> >
> > [3] Norman et al. "Exploring genetic interaction manifolds constructed from rich single-cell phenotypes" Science (2019)
> >
> > [4] Adamson et al. "A Multiplexed Single-Cell CRISPR Screening
> > Platform Enables Systematic Dissection of the
> > Unfolded Protein Response" Cell (2016)
> >
> > [5] Heimberg et al. "Low Dimensionality in Gene Expression Data Enables the Accurate Extraction of Transcriptional Programs from Shallow Sequencing" Cell Systems (2016)
> >
> > [6] Cleary et al. "Efficient Generation of Transcriptomic Profiles by Random Composite Measurements" Cell (2017)

---

### Official Review · Reviewer_AHb3 · 2022-07-10

**Rating:** 7
**Confidence:** 4
**Soundness:** 3 good
**Presentation:** 3 good
**Contribution:** 4 excellent

**Summary:**

This paper proposes an approach to causal discovery in high dimensional settings where interventional data is available. The authors use a low-rank assumption on a factor graph, and incorporate the model into a differentiable causal discovery algorithm by utilizing a likelihood model which assumes perfect interventions. Empirical results show strong performance with respect to prior art.

**Questions:**

* Is there a parameterization that would allow for the rank to be chosen as well? Perhaps I am missing something along these lines.
* The work assumes perfect interventions in the likelihood. Would this line of work extend to imperfect interventions? Purely observational settings? It would be nice if the authors were able to clearly delineate within the text where the interventions are strictly required.
* Is it possible to have a set of experiments, as detailed above, that show robustness to misspecification?

**Limitations:**

The authors do a nice job of describing their work within the parameters of necessary assumptions and constraints. I don't foresee clear societal impact issues ehre.

**Strengths And Weaknesses:**

Strengths:

* The introduction of a method that allows for a low rank factor assumption is very nice, and provides a valuable tool in causal discovery where sparsity can be a difficult assumption to enforce.
* The approach is a nice and relatively simple solution to a difficult problem.
* The parameterization of the model with the Gumbel-softmax prior is quite nice.

Weaknesses:

* Unclear how the number of factors should be chosen.
* Experimental evidence which shows the robustness to misspecification (both in terms of rank and the Gaussian assumption) would be helpful.
* (Minor) It would be good if the authors more clearly delineated that this paper deals solely with interventional data.

---

> ### Author Response · Authors · 2022-08-02
> **Response to reviewer**
>
> We thank the reviewer for their review and are glad they appreciated the simplicity of our proposed algorithm.
>
> **Choice of number of factors:** We agree with the reviewer that choosing the number of factors is an important consideration. However, having at least an upper bound on the number of factors is crucial to guarantee a reasonable runtime of DCD-FG. Importantly, as long as the number of factors needed for good statistical performance is not too large, we found that a good number can be easily found by optimizing the negative log-likelihood on a validation dataset, which is how we performed the reported experiments. Such a procedure is already necessary for searching for the strength of sparsity regularization in DCDI / NOTEARS, and therefore introduces minimal overhead.
>
> More specifically, we investigated the performance of DCD-FG for different values of the number of factors ($m$) in the simulated data from the linear causal mecanism. For several values of $m$, we selected the best performing model based on cross-validation across other hyperparameter (e.g., sparsity), but only for that specific $m$ (DCD-FG [$m$=X]). We reported the performance of this model (F1-score), and compared it to the best performing model across the whole hyperparameter grid (DCD-FG). We observe that performance is better for $m=15$, but does not drastically change for values of $m$ close to the groundtruth. Second, we verified that the performance in the hyperparameter grid (here denoted by F1 score), was well correlated with the validation likelihood. All in all, those results suggest that $m$ can be effectively chosen via cross-validation. We will add those results into new figure to the final version of the paper (https://anonymous.4open.science/r/rebuttal-neurips-D93C/rank-score.pdf).
>
> There may be opportunities for extensions with adaptive / non-parametric number of factors, but we leave this for future work.
>
>
> **Other interventional settings:** We envision that our framework should be compatible with extensions to imperfect interventions or interventions with unknown targets, similar to those incorporated into DCDI [1]. However, the factor semantics might slightly complicate these extensions, and we consider them out of the scope of our current work. They are definitely exciting venues to be explored in future work, as we will note in the revised discussion.
>
> Our framework technically does apply to observational data, but we would argue that the signal-to-noise ratio of such data, especially in the biological applications that we highlight, is too low for any reasonable inference, so we would consider it out of scope for the current publication. Moreover, given recent substantial advances in experimental biology, interventional data is rapidly growing, both in lab models, and in human data (with natural genetic variants as interventions). This offers an important -- and currently underserved -- use case. We will clarify this in the camera-ready version of the paper.
>
> **Robustness to misspecification:** As mentioned above, we actually chose the rank in our experiments adaptively, and the best performing rank was usually slightly higher than the ground truth rank. Thus, we concluded that DCD-FG is not sensitive to the exact ground truth rank.
>
> To address robustness to misspecification of the noise model, we will add new experiments to the Supplementary Materials. For these experiments, we repeated the experiments from Figure 4 of the paper, but changed the Gaussian noise to uniform noise, with matching variance. We report the interventional likelihood (I-NLL), as well as the F1 score for the top three performing methods, NOTEARS, NOTEARS-LR, and DCD-FG. The results of these experiments can be found here: https://anonymous.4open.science/r/rebuttal-neurips-D93C/linearuniform.pdf & https://anonymous.4open.science/r/rebuttal-neurips-D93C/nnuniform.pdf. Note that the F1 scores are slightly lower than for the Gaussian simulations, presumably because of the effects of model misspecification. However, DCD-FG clearly outperforms NOTEARS and NOTEARS-LR as in our previous experiments.
>
> [1] Brouillard, Philippe, et al. "Differentiable causal discovery from interventional data." Advances in Neural Information Processing Systems 33 (2020): 21865-21877.

---

### Official Review · Reviewer_ADLi · 2022-07-11

**Rating:** 5
**Confidence:** 5
**Soundness:** 2 fair
**Presentation:** 3 good
**Contribution:** 2 fair

**Summary:**

The paper proposed a new DAG constraints for low rank adjacency matrices. Compared to original DAG constraints, the new constraint can scale to larger graphs. On binary graph, the proposed DAG constraints is a necessary and sufficient condition for an adjacency matrix to form a DAG. However, for a weighted adjacency matrix, it seems that without further work, the proposed DAG constraint is a sufficient but not necessary condition. In that case the proposed method actually is optimising over a space that is smaller than the true DAG space.

**Questions:**

See my comments above.

**Ethics Review Area:**

["I don’t know"]

**Limitations:**

1. The main result is trivial and it is only an sufficient condition of DAG.
2. There may be no real identifiability result.

**Strengths And Weaknesses:**

The idea of the paper is very similar to [1], and [1] is rejected by ICLR2020 and now it is only a preprint. Unfortunately, the paper share the same fatal problem as [1]. Let us consider an adjacency matrix W = UV, if we enforce that
When apply such low rank DAG constraints, it is easy to derive a necessary and sufficient condition on binary adjacency matrices. If W is a weighted matrix, then we have to first make the entries of W to be positive to apply the NOTEARS type DAG constraints. In this procedure, in order to use the low rank property, one has to apply absolute value, or square over entries of U and V. In that case, the DAG constraint becomes sufficient only.

The main result (Proposition 2) of the paper is actually trivial. The proof in the supplementary is over complicated. In fact the proof of this trivial result is very straightforward. For a matrix W \in R_{>=0}^{n\times n} = UV, where all entries of U and V are also no-negative, W is nilpotent if and only W is a DAG. Furthermore, VU is nilpotent if and only if UV is nilpotent, and thus VU is DAG is equivalent to UV is DAG.

There is also a minor problem. The rank of DAG graphs can be very large. For example, if we consider a chain structured DAG with n nodes, the rank of its adjacency matrix is n - 1. Thus the proposed method only suits sparse graph structures.

Finally, in the identifiability section, the authors provide a theorem about the rank of the graph? How the the theorem related to the identifiability? Will this enforce the faithfulness of the discovered graph?
[1] Fang, Zhuangyan, et al. "Low rank directed acyclic graphs and causal structure learning." arXiv preprint arXiv:2006.05691 (2020).

---

> ### Author Response · Authors · 2022-08-02
> **Response to reviewer (1/3)**
>
> We thank the reviewer for their assessment, in particular for acknowledging the beneficial scaling of our proposed method. First, we clarify that we consider the scope of our paper as more comprehensive than just introducing new DAG constraints. We introduce novel, nested complexity classes of structural equation models that encompass both the DAG structure and the functional relationships between variables. Both are crucial to enable scaling of the causal discovery method to thousands of variables.
>
> In the following, we address specific points raised by the reviewer.
>
> **Necessary vs sufficient DAG conditions** We argue that the criticism pointed out by the reviewer does not apply to neither of our work, and of the previous low-rank work.
>
> To demonstrate this, we would like to start by distinguishing between several classes of low-rank graphs on $d$ nodes:
>
> \begin{align*}
> \mathcal{G}_{\mathrm{lin}}^m &= \\{ G : G \text{ admits weighted adjacency matrix}~ W \in \mathbb{R}^{d \times d} \\\\
> & \quad \quad \text{ with } W=UV \text{ for } U \in \mathbb{R}^{d \times m}, V \in \mathbb{R}^{m \times d}\\}, \text{ the set of linear rank $\leq m$ graphs}
> \end{align*}
>
> \begin{align*}
> \mathcal{G}_{\mathrm{lin,nonneg}}^m &= \\{ G : G \text{ admits weighted adjacency matrix } W \in \mathbb{R}\_{\geq 0}^{d \times d} \\\\
> & \quad \quad \text{ with } W=UV \text{ for } U \in \mathbb{R}\_{\geq 0}^{d \times m}, V \in \mathbb{R}\_{\geq 0}^{m \times d}\\}, \text{ the set of linear non-negative rank $\leq m$ graphs}
> \end{align*}
>
> \begin{align*}
> \mathcal{G}_{\mathrm{bool}}^m &= \\{ G : G \text{ has adjacency matrix } A \in \\{0,1\\}^{d \times d} \\\\
> & \quad \quad \text{ with } A=U \diamond V \text{ for } U \in \\{0,1\\}^{d \times m}, V \in \\{0,1\\}^{m \times d}\\}, \text{ the set of Boolean rank $\leq m$ graphs}
> \end{align*}
>
>
>
> Note that in the definition of $\mathcal{G}\_{\mathrm{lin}}^m$, we allow for $W$ to encode the presence of an edge in $G$ with any non-zero entry, positive or negative. One can easily show that $\mathcal{G}\_{\mathrm{bool}}^m = \mathcal{G}\_{\mathrm{lin,nonneg}}^m \subsetneq \mathcal{G}_{\mathrm{lin}}^m \subsetneq \mathcal{G}$ for $m < d$.
>
> We point out a subtlety here: for a given matrix, its non-negative rank and Boolean rank do not necessarily coincide, but $\mathcal{G}\_{\mathrm{bool}}^m = \mathcal{G}\_{\mathrm{lin,nonneg}}^m$ since we allow for arbitrary weighted adjacency matrices in the definition of $\mathcal{G}\_{\mathrm{lin,nonneg}}^m$.
>
> The paper mentioned by the reviewer [1] in fact deals with $\mathcal{G}\_{\mathrm{lin}}^m$ and considers two strategies for enforcing this low rank constraint (explicit parameterization and nuclear norm penalization). In both cases, the DAG constraint is simply enforced by first expanding the full adjacency matrix $W$ and then enforcing $h_d(\mathrm{abs}(W))=0$. Consequently, this methodology does not require to apply the absolute value to each of the matrices $U$ and $V$, and seems to not have the flaw that the reviewer pointed out: their strategy searches through all DAGs in $\mathcal{G}_{\mathrm{lin}}^m$. However, this design also precludes [1] from harnessing any benefit in asymptotic algorithmic complexity.
>
> By contrast, we do _not_ consider $\mathcal{G}\_{\mathrm{lin}}^m$, but instead choose to exclusively work with $\mathcal{G}\_{\mathrm{bool}}^m$. We enforce this constraint by searching over non-negative decompositions of the weighted adjacency matrix (obtained by the Gumbel-sigmoid parametrization). In fact, we make use of the following slight extension of Proposition 7 in our Appendix: If $G \in \mathcal{G}^m\_{\mathrm{bool}}$, then
>
> $$
> \begin{alignat*}{2}
> G \text{ is DAG} \Leftrightarrow & \exists U \in \mathbb{R}\_{\geq 0}^{d \times m}, V \in \mathbb{R}\_{\geq 0}^{m \times d}: {}&& W = UV \text{ is weighted adjacency matrix for } G \\
> &&& \text{ with } h_d(UV) = 0\\
> \Leftrightarrow & \exists U \in \mathbb{R}\_{\geq 0}^{d \times m}, V \in \mathbb{R}\_{\geq 0}^{m \times d}: && {} W = UV \text{ is weighted adjacency matrix for } G\\
> &&&\text{ with } h_m(VU) = 0
> \end{alignat*}
> $$
>
> In effect, in our paper, we search over all DAGs contained in $\mathcal{G}^m_{\mathrm{bool}}$ since the continuous relaxation we employ is indeed a necessary and sufficient condition for $G$ to be a DAG. As we show in our paper, considering $\mathcal{G}^m_{\mathrm{bool}}$ instead of $\mathcal{G}^m_{\mathrm{lin}}$ gives further rise to an intuitive way of restricting the nonlinear _functional_ relationships on top of the graphical structure while maintaining low asymptotic computational complexity. This aspect is completely absent from [1].

---

> > ### Author Response · Authors · 2022-08-02
> > **Response to reviewer (2/3)**
> >
> > We believe the reviewer is concerned that our method does not search the entire space of linear low-rank graphs, $\mathcal{G}\_{\mathrm{lin}}^m$. Indeed, by the strict inclusion $\mathcal{G}\_{\mathrm{bool}}^m \subsetneq \mathcal{G}\_{\mathrm{lin}}^m$, finding a decomposition of the weighted adjacency matrix $W=UV$ with non-negative $U \in \mathbb{R}\_{\geq 0}^{d \times m}, V \in \mathbb{R}\_{\geq 0}^{m \times d}$ is only sufficient for $G \in \mathcal{G}\_{\mathrm{lin}}^m$ to be a DAG, but not necessary, since we could be missing graphs in $\mathcal{G}\_{\mathrm{lin}}^m \setminus \mathcal{G}\_{\mathrm{bool}}^m$. Similarly, assume we start with an arbitrary linear decomposition with no sign constraints on $W$, $U$, and $V$, i.e., $W = UV$, $U \in \mathbb{R}^{d \times m}, V \in \mathbb{R}^{m \times d}$. In order to check whether $W$ corresponds to a DAG, we need to apply a transformation to $W$ that turns its negative entries into positive ones, for example taking the element-wise absolute value $\mathrm{abs}(W)$. This precludes us from applying the trick in Proposition 7, namely, $h_d(\mathrm{abs}(UV)) \neq h_m(\mathrm{abs}(VU))$.
> >
> > However, this is only a concern if there is reason to believe that precisely searching through DAGs in $\mathcal{G}\_{\mathrm{lin}}^m$ for our dependency masks is of any particular practical interest. Besides being more immediate when starting from a linear structural equation model, we see no particular reason for favoring $\mathcal{G}\_{\mathrm{lin}}^m$ over $\mathcal{G}\_{\mathrm{bool}}^m$ in light of the practical benefits outlined in our paper. Moreover, the only linear low-rank models not captured in $\mathcal{G}\_{\mathrm{bool}}^m$ are those in which the contributions of multiple factors cancel out to produce more zeros than expected from the sparsity pattern of the factors $U, V$, corresponding to a lack of faithfulness of the factor graph. We consider these models edge cases that could safely be excluded from the search space.
> >
> > To address these issues in the revised manuscript, we will add comments clarifying the relationship between these different sets of graphs to the supplementary material of the paper and refer to them from the main text.
> >
> > If we misunderstood the reviewer's concerns, we would appreciate some further clarification from the reviewer to help us address the points raised.
> >
> > **Similarities to Fang, Zhuangyan, et al.:** The reviewer raises concerns about similarity in our results with [1], and we would like to summarize the key similarities and differences between the two works here. Both [1] and our work employ low-rank assumptions in the context of continuous optimization-based DAG learning algorithms. However, our work differs from [1] in several key aspects:
> >
> > 1. In the above notation, [1] consider the class of **linear** low-rank graphs $\mathcal{G}\_{\mathrm{lin}}^d$ (potentially given by the coefficients of a linear Gaussian SEM) while we focus on Boolean low-rank $\mathcal{G}\_{\mathrm{bool}}^d$ graphs. Our work is more general and separates the inference of the graph, and of the likelihood model, and makes a distinction between the matrix rank and the Boolean rank. In particular, this allows us to not only exploit the rank constraint for the purpose of the penalty evaluation, but also to consider a novel class of functional relationships with beneficial effects on statistical and computational performance.
> > 2. [1] barely exploits the low-rank assumption for computational gains and they only provide a very cursory treatment of runtime issue. In fact, as explained above, to the best of our understanding, all algorithms suggested in [1] explicitly expand the full (weighted) adjacency matrix before applying the trace exponential penalty, therefore incurring an asymptotic complexity of $O(d^3)$ per penalty evaluation. Only in the linear case do they exploit an explicit low-rank parametrization that lowers the time complexity for the fitting term evaluation. Indeed, we could find a table that recapitulates walltime to run their algorithms with different parameters on simulated data, which shows that NOTEARS-LR may be twice as slow as NOTEARS for 300 nodes and a rank of 30. More importantly, the low-rank version of GraN-DAG has similar or higher runtime complexity (due to the calculation of the nuclear norm) compared to GraN-DAG. Because GraN-DAG requires one neural network per node (as DCDI), this regularization approach is not applicable to thousands of variables (as DCDI). By contrast, we discuss time and space complexity extensively in the paper, and empirically compare runtime per iteration. In all cases, we show significant improvement compared to DCDI / NOTEARS through **cleanly exploiting the low-rank constraint** on the graph for both the penalty and the fitting term.
> >
> > We will add these comments as an additional discussion in the supplementary materials of the paper.

---

> > > ### Author Response · Authors · 2022-08-02
> > > **Response to Reviewer (3/3)**
> > >
> > > **Triviality of Proposition 2:** We thank the reviewer for pointing out this algebraic argument for proving part of Proposition 2. We would like to make a few additional comments on this point:
> > > 1. We think that the elementary nature of the proof we provide for Proposition 2 has its merits. Indeed, it is based on an elementary graphical argument (the definition of a cycle, and a path) and does not require any knowledge of linear algebra. Moreover, we do not consider it significantly longer or more complicated than the proof provided by the reviewer.
> > > 2. While we admit it is a simple result and the reviewer considers it obvious, the paper cited by the reviewer, [1], did _not_ exploit this result in a similar context, leading to a less efficient algorithm.
> > > 3. We do not consider the proof of Proposition 2 the main result of our paper but rather part of a larger, novel framework of low-rank assumptions for causal discovery, together with further implementation details such as the functional form of the structural equations. In terms of our theoretical contributions, we also highlight Theorem 1 as one of the main results.
> > >
> > > Based on the relative simplicity of the proof, we will change the designation of Proposition 2 to be a Lemma in the revised manuscript. We will also add the elegant, alternative algebraic proof provided by the reviewer to the relevant section of the paper.
> > >
> > > **Expressivity of low-rank factor graphs:** We agree with the reviewer that general DAGs cannot be represented as the half-square of a factor graph with a low number of factors. The chain structure is one instance, but in general, Erdos-Renyi graphs may also have high (matrix and likely Boolean) rank, as explained in [1]. However, our work is very motivated by biological applications, in which many genes are affected by similar other genes, forming a very dense interconnected network with several core sets of gene regulators. In this setting, using factor DAGs is an extremely appealing modeling technique. We will clarify this motivation in the revised manuscript.
> > >
> > > Moreover, it is evident from our experiments that the space of all DAGs together with arbitrary functional relationships might simply be too large to be learned efficiently when the number of nodes is high, necessitating _some_ form of statistical regularization. Simple sparsity regularization, as provided by NOTEARS, did not seem sufficient. Thus, we argue that factor graphs provide one possible route of extracting some meaningful causal relationships, even without exhaustively covering the space of all DAGs.
> > >
> > > We will also clarify the limited expressivity of factor graphs in the discussion section, focusing on the rationale for using them in the context of biological data.
> > >
> > > **Theorem 1 (rank increase) in identifiability section, and consequences on faithfulness:** We apologize for any confusion regarding the relationship of Theorem 1 with the identifiability of the underlying graph. Let us rephrase the key results of Section 3.2:
> > >
> > > When investigating the identifiability of the underlying DAG in the infinite data limit, one can appeal to standard results for causal discovery, such as those employed in [2], to show that the skeleton of the graph can be recovered _if_ the underlying structural equation model is faithful. Note that faithfulness here is not simply a property of the graph, but of the full probabilistic model. Since the classes of graphs we consider are highly connected, for random graphs, we show in Appendix B (see Lemma 2 and Proposition 6) that recovering the skeleton in turn is sufficient to identify the whole graph.
> > >
> > > Next, we argue that identifiability is a very low bar to clear, given how far we can be in practice from the "infinite data" regime. Thus, we set out to investigate other notions of statistical sensitivity of the problem in question. We assumed that the lack of data could easily lead to errors in the recovery of the skeleton of the ground truth graph. Thus, we proceeded to show in Theorem 1 that (random) corruptions in the graph lead to a (Boolean) rank increase with high probability. This provides evidence for the beneficial statistical effect of restricting the (Boolean) rank of the graphs in our search space.
> > >
> > > In order to clarify our reasoning to the reader, we will edit the relevant section of the paper, in particular changing its title from "Identifiability of Random Causal Factor Graphs" to "Statistical Properties of Random Causal Factor Graphs".
> > >
> > >
> > > [1] Fang, Zhuangyan, et al. "Low rank directed acyclic graphs and causal structure learning." arXiv preprint arXiv:2006.05691 (2020)
> > > [2] Brouillard, Philippe, et al. "Differentiable causal discovery from interventional data." Advances in Neural Information Processing Systems 33 (2020): 21865-21877.

---

> > > > ### Comment · Reviewer_ADLi · 2022-08-08
> > > > **Identifiability**
> > > >
> > > > I agree with the author that if we decompose the graph structure and the parameter, then DAG constraint becomes sufficient and necessary.
> > > >
> > > > I just noticed the paper are working on interventional data. Meanwhile, the baselines, notears and notears-lowrank are developed for observational data. Thus it would be good if there is some results on observational data.

---

> > > > > ### Author Response · Authors · 2022-08-08
> > > > > **Response to reviewer's comment**
> > > > >
> > > > > We thank the reviewer for acknowledging that in the context of our paper, the DAG constraints we consider are both necessary and sufficient.
> > > > >
> > > > > The reviewer raises the point that the subset of baselines derived from NOTEARS might not be adequate in this benchmarking because they have been designed for inference on observational data. First, we would like to highlight that IGSP is specifically designed to handle interventional data, and has been part of our comparisons throughout all of our experimental settings. Second, while we agree with the reviewer that NOTEARS has originally been proposed for observational data, we believe that our adaptation of NOTEARS and NOTEARS-LR to the (perfect) interventional setting is a sensible point of comparison (full description in Appendix F.3). Indeed, this extension simply consists in masking of likelihood for the intervened nodes as proposed in the DCDI manuscript [1]. In fact, similar adaptations of NOTEARS have already been considered in the context of learning from temporal data [2], so we would consider them fairly standard by now. Additionally, we would like to note that the NOTEARS-LR extension performs extremely well on the linear Gaussian SEM experiments (Figure 4).
> > > > >
> > > > > Then, the reviewer proposes that we apply the method in the purely observational setting for fair comparison with NOTEARS. Overall, we consider this investigation to be out of scope for this manuscript for the following reasons:
> > > > > 1. The observed beneficial performance of NOTEARS on observational data might be due to specific properties of the simulated data [3, 4] exploited during inference time. Indeed, the authors of those papers showed that z-scoring the data before feeding it to NOTEARS caused it to perform as poorly as random guessing. However, in this work and in the DCDI [1] paper, we present experiments with interventional data in which gradient-based methods lead to state-of-the-art results *after data normalization*.
> > > > > 2. The signal-to-noise ratio of observational data, especially in the biological applications that we highlight, is too low for any reasonable inference, as has been highlighted in the field of single-cell RNA-seq [5].
> > > > > 3. Given recent substantial advances in experimental biology, interventional data is rapidly growing, both in lab models, and in human data (with natural genetic variants as interventions). Our work offers an important – and currently underserved – use case.
> > > > > We will add these discussions in the camera-ready version of the paper.
> > > > >
> > > > > We also will consider adding some benchmarks in the observational setting to the final version of the paper, but given the current time-constraints (~20 hours left until the end of the author-reviewer discussion period), it is unlikely that we will be able to accommodate this additional request for benchmarks during this present discussion period.
> > > > >
> > > > > [1] Brouillard, Philippe, et al. "Differentiable causal discovery from interventional data." Advances in Neural Information Processing Systems 33 (2020): 21865-21877.
> > > > > [2] Gao, Tian, et al. "IDYNO: Learning Nonparametric DAGs from Interventional Dynamic Data." International Conference on Machine Learning. PMLR, 2022.
> > > > > [3] Reisach, Alexander Gilbert, et al. "Beware of the Simulated DAG! Causal Discovery Benchmarks May Be Easy to Game." Advances in Neural Information Processing Systems 34 (2021)
> > > > > [4] Kaiser, Marcus, et al. "Unsuitability of NOTEARS for Causal Graph Discovery." arXiv (2021)
> > > > > [5] Pratapa, Aditya, et al. "Benchmarking algorithms for gene regulatory network inference from single-cell transcriptomic data." Nature Methods 17-147–154 (2020)

---

> > > > > > ### Author Response · Authors · 2022-08-09
> > > > > > **Supplementary Experiments**
> > > > > >
> > > > > > Although we leave the development of thorough benchmarking studies in the context of observational data as future work, we are happy to report that we managed to run a limited set of experiments to accommodate the reviewer's request.
> > > > > >
> > > > > > More precisely, we generated data in two experimental settings (linear and non-linear), matching the parameters we use in the paper (d=100, m=10), but with only observational data. Because of time constraints, we simulated data for only five sampled DAGs per setting instead of ten. We report results for NOTEARS, NOTEARS-LR, as well as DCD-FG (the top three performing methods throughout the paper).
> > > > > >
> > > > > > The results of these experiments (https://drive.google.com/file/d/1PlocBals72tAhSeZ-ShFdU6yRooT8GsJ/view?usp=sharing; files nlin-no-interv.pdf and lin-no-interv.pdf)** show that performance is overall lower on observational data compared to interventional data. This is expected as each intervention provides richer information about the causal structure, and similar results are reported in the appendices of the DCDI paper. However, DCD-FG still outperforms the other competing methods. Especially, it is interesting that DCD-FG outperforms NOTEARS-LR by a large margin, even on the linear dataset.
> > > > > >
> > > > > > We will add these results to the final version of the paper. However, even though the results suggest that the method is largely applicable on observational data, we will still carefully discuss that the scope of the paper is only interventional data.
> > > > > >
> > > > > > ** Unfortunately anon4openscience is currently down at the time of writing this post, so we have uploaded the results onto another public hosting service (Google Drive). We will attempt tomorrow posting them to anon4openscience as well.

---

### Author Response · Authors · 2022-08-01
**General Response**

We thank the reviewers for their valuable feedback. Reviewer AHb3 concisely conveyed the intuition of our contribution: “the authors [...] introduce factor directed acyclic graphs (f-DAGs) which have many promising properties by incorporating the structural assumption to restrict the search space and reduce the optimization difficulty”. All reviewers point out the scalability of our approach: "Compared to original DAG constraints, the new constraint can scale to larger graphs" (R ADLi). Two out of three reviewers explicitly highlight the soundness of the method, and the strength of our empirical evaluation. For example, the reviewers mention that "The approach is a nice and relatively simple solution to a difficult problem. [...] Empirical results show strong performance with respect to prior art. (R AHb3), "Theoretical analyses and experimental results demonstrate the superiority of f-DAGs and DCD-FG. [...] The experiments are well-designed." (R WAgt).

Reviewer ADLi has questions about the potential restrictiveness of the class of low-rank graphs we consider for DCD-FG (Boolean low-rank), and why we supposedly discard the treatment of more general scenarios (matrix low-rank). We address these questions in this response, and will include them in the camera-ready version. In a nutshell, we argue that matrix low-rank makes sense for the special case of linear SEMs, while our class of Boolean low-rank graphs is particularly meaningful for reasoning about non-linear causal models, which is the focus for this paper, and is especially important in some domains, such as biological genetic networks.

We address the concerns of each reviewer in the point by point response below. In some of the answers, we include results from novel experiments we performed in response to the reviewers' questions and suggestions.

---

### Author Response · Authors · 2022-08-09
**Filehosting service Anonymous4OpenScience down, alternative link provided**

Dear reviewers and area chairs,

We noticed that the server of anonymous4openscience was down, so we have took the liberty to upload our files to a separate Google Drive so that the remaining reviewer(s) can access the results of our supplementary experiments.

https://drive.google.com/file/d/1PlocBals72tAhSeZ-ShFdU6yRooT8GsJ/view?usp=sharing

Thanks for your attention,
The Authors,

---

### Meta-Review · Area_Chair_VCc7 · 2022-08-26

**Recommendation:** Accept
**Confidence:** Certain

**Metareview:**

In this paper, the authors propose a new DAG constraint for low-rank adjacency matrices., which can scale to larger graphs. All the reviewers consider this paper is sound and the experiments are well designed. However, one question about the case of different graph spaces from other reviewer should be addressed in the final version.

**Award:**

No

---

### Decision · Program_Chairs · 2022-09-14

Accept